# Message Passing on the Edge: Towards Scalable and Expressive GNNs

**Pablo Barceló** [1 2 3]   **Fabian Jogl** [4]   **Alexander Kozachinskiy** [2]   **Matthias Lanzinger** [4]   **Stefan Neumann** [4]
**Cristóbal Rojas** [1 2]

## Abstract

Graph neural networks (GNNs) are widely used in graph learning and most architectures propagate information by passing messages between vertices. In this work, we shift our attention to GNNs that perform message passing on *edges* and introduce EB-1WL, an edge-based color-refinement test, and a corresponding architecture, EB-GNN. Our EB-GNN architecture is inspired by the classic triangle-counting algorithm of Chiba and Nishizeki and passes messages along edges and triangles. Our contributions are as follows: (1) Theoretically, we show that EB-1WL is significantly more expressive than 1WL. We provide a complete logical characterization of EB-1WL in first-order logic, along with distinguishability results via homomorphism counting. To the best of our knowledge, EB-GNN has the strongest theoretical expressivity guarantees among edge-based message-passing GNNs in the literature. (2) Unlike many GNN architectures that are more expressive than 1WL, we prove that EB-1WL and EB-GNN admit near-linear time and memory usage on practical graph learning workloads. (3) We show in experiments that EB-GNN is a highly efficient general-purpose architecture: it substantially outperforms simple MPNNs and remains competitive with task-specialized state-of-the-art GNNs at substantially lower computational cost.

## 1. Introduction

Graph neural networks (GNNs) have emerged as a fundamental tool across the sciences (Scarselli et al., 2009; Kipf & Welling, 2017; Wu et al., 2019). To explain the success and

[1]IMC, Pontificia Universidad Católica de Chile, Santiago, Chile [2]CENIA, Santiago, Chile [3]IMFD, Santiago, Chile [4]TU Wien, Vienna, Austria. Correspondence to: Fabian Jogl <fabian.jogl@tuwien.ac.at>, Matthias Lanzinger <matthias.lanzinger@tuwien.ac.at>.

*Proceedings of the 43rd International Conference on Machine Learning*, Seoul, South Korea. PMLR 306, 2026. Copyright 2026 by the author(s).

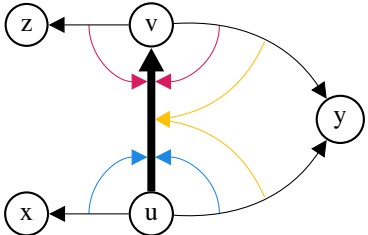

*Figure 1.* Message passing scheme of EB-1WL and EB-GNN. Colors correspond to aggregation types.

limitations of GNNs, a lot of research has been dedicated to understanding the expressiveness of GNN architectures. While the most prominent approach is the Weisfeiler–Leman test (Xu et al., 2019; Morris et al., 2019), several alternative, yet equally insightful, characterizations of GNN expressive power have been developed. Particularly, researchers also obtained characterizations of GNN expressiveness through finite-variable fragments of counting logics or homomorphism counts from classes of graphs of bounded treewidth (Dvořák, 2010; Dell et al., 2018).

A key limitation of existing expressivity results is that 1WL, despite its computational efficiency, fails to capture important motifs such as triangles and other substructures (Chen et al., 2020; Arvind et al., 2020; Lanzinger & Barceló, 2024). The higher-order WL hierarchy offers a principled way to overcome these limitations: properties invisible to 1WL are often detected by $k$WL for some $k > 1$. However, even modest increases in expressivity quickly become computationally prohibitive—already 2WL-inspired GNNs require quadratic memory and cubic time in the number of nodes, making them impractical for large graphs (Maron et al., 2019). This leads to a central question: *Can we go beyond 1WL while retaining near-linear cost on sparse graphs?*

Additionally, most of the aforementioned expressivity results have been derived for GNNs which perform message passing on a graph's vertices. However, recently it has been observed that in certain application domains, such as chemistry, GNNs that pass messages along the *edges* tend to outperform their vertex-centric competitors (Jian et al., 2025; Vadaddi et al., 2024; Song et al., 2020; Heid et al., 2024; Heid & Green, 2022). Thus, another important challenge is as follows: *Can we obtain strong expressivity*

*guarantees for GNNs with edge-based message passing?*

**Our contributions.** We provide clear positive answers to both of these research questions. We introduce a novel GNN architecture which performs edge-based message passing and which explicitly takes into account triangles to achieve improved expressivity. We prove strong theoretical expressivity results, showing that our architecture is more expressive than 1WL and providing a logical characterization through first-order logic and homomorphism counts. We further show that our architecture requires only near-linear time in practice. We are not aware of any other edge-based message-passing GNNs in the literature with such clear understanding of expressivity and strong theoretical guarantees.

More concretely, we introduce the *edge-based* 1WL test (EB-1WL) — a color-refinement test that is provably more expressive than 1WL while still being computationally efficient. EB-1WL updates edge colors through incidence relations and triangle-induced interactions (see Fig. 1 and Sec. 3), an edge-centric perspective inspired by the classic triangle-counting algorithm of Chiba & Nishizeki (1985). This approach retains much of 2WL's relational strength while avoiding its combinatorial blowup: each iteration runs in near-linear time on sparse graphs. Concretely, each iteration takes $O(\alpha m)$ time, where $m$ is the number of edges and $\alpha$ is the graph's arboricity. Since $\alpha$ is typically small in real-world graphs,[1] EB-1WL achieves near-linear performance in practice.

We further prove three expressiveness results for EB-1WL:

1. EB-1WL strictly extends the expressive power of 1WL and of NC-1WL by Liu et al. (2024).
2. On the logical side, EB-1WL admits a precise characterization in terms of clique-based finite-variable fragments with counting quantifiers.
3. On the homomorphism-count side, we show that EB-1WL is at least as powerful as counting homomorphisms from chordal graphs of treewidth two.

Building on this foundation, we introduce EB-GNNs, an edge-based message-passing architecture that exactly matches the expressive power of EB-1WL. In experiments, EB-GNNs perform strongly across diverse settings:

1. On synthetic benchmarks, EB-GNNs capture structural patterns beyond triangle counts and, in some cases, rival higher-order tests.
2. On molecular datasets, they outperform standard baselines and remain competitive with task-specialized GNNs

while being significantly more computationally efficient.
3. On large-scale graphs, they scale effectively and achieve state-of-the-art accuracy.

Together, these results establish EB-GNNs as an expressive, yet highly efficient general-purpose architecture.

**Related work.** Surprisingly, *edge-based GNN architectures* are rare in the literature. Recent work by Liu et al. (2024) introduced the *neighbor-communication* 1WL (NC-1WL) test, extending 1WL by incorporating edge information within the neighborhood of each node in the graph. They show that NC-1WL is strictly more expressive than 1WL and remains efficient, making it an attractive refinement from a practical standpoint. However, from a theoretical perspective, its study remains incomplete: while NC-1WL has a well-defined placement within the WL hierarchy, no alternative characterizations (such as logical or homomorphism-count based) are known. Here, we show that our proposed EB-1WL is more expressive than NC-1WL and that in experiments our methods achieve better results. Additionally, Zhang et al. (2020) propose an architecture based on edge convolution. Cai et al. (2022) studied the application of standard message-passing GNNs to the line graph[2] for link prediction.

Another line of related work concerns the study of *efficient higher-order GNNs* which is typically conducted for vertex-based message passing. Here various approaches have been proposed to improve on higher-order $k$-WL performance bottlenecks especially for sparse graphs. Morris et al. (2020) introduced $\delta$-$k$-WL and its local variants, which demonstrate improvements on sparse graphs while maintaining high expressivity. However, it remains rather computationally inefficient, as it mimics the non-folklore versions of the higher-order WL test. While improving upon them in the aggregation time by considering only local interactions, it still requires $n^3$-space to get above the 1-WL's expressive power. Zhao et al. (2022) follow a similar motivation as Morris et al. (2020) and introduced $(k, c)(\leq)$-SetWL, which approximates higher-order power in a fine-grained way by also embedding local substructures. While these refinements are more scalable than plain 2WL, the vertex-based paradigm they follow makes quadratic (in the number of vertices) running time unavoidable, even in sparse graphs.[3] Moreover, such practical considerations remove these methods from the strong theoretical foundations of $k$-WL, which admits well-known and highly influential characterizations in terms

---

[1]In all 39 datasets considered by Eppstein et al. (2013) the arboricity is at most 201 even though their largest graph has 3.7 million nodes and 16.5 million edges. On 32/39 of their datasets, the arboricity is less than 60. We note that Eppstein et al. (2013) report the degeneracy, which is an upper bound on the arboricity.

[2]In the *line graph* of a graph $G$, the edges of $G$ become vertices that are connected according to their incidences in $G$.

[3]The closest comparison to our method in terms of $(k, c)(\leq)$-SetWL is achieved with $k = 3, c = 1$ (lower $k$ yields expressivity at most 1WL). On a star graph this materializes all quadratically many 2-hop paths as nodes in a "super-graph" on which message-passing is performed. In contrast, a star has arboricity 1 and our method is strictly linear in its running time.

of homomorphism counts (Dvořák, 2010; Dell et al., 2018) and variants of first-order logic (Cai et al., 1992). Along the same lines, other prominent approaches to obtain efficient higher-order GNNs such as PPGN (Maron et al., 2019) also cannot avoid fundamentally quadratic (or higher) computational time complexity per layer.

## 2. Preliminaries

*Graphs.* We study undirected graphs $G = (V, E)$, where $V$ is the set of vertices and $E \subseteq \binom{V}{2}$ is the set of edges. We set $n = |V|$ and $m = |E|$. We write $N(v) := \{w \mid \{v, w\} \in E\}$ for the *neighborhood* of $v \in V$. For technical simplicity, we assume that graphs do not have isolated nodes.

The *arboricity* $\alpha$ of a graph $G = (V, E)$ is the minimum number of forests that partition its edge set $E$ (Diestel, 2012). More formally, the arboricity is the smallest integer of $k$ such that there exist forests $F_1, \ldots, F_k$ with $F_i = (V, E_i)$ such that $\bigcup_{i=1}^{k} E_i = E$. The arboricity is also tightly related to other important graph parameters, such as the *degeneracy* or the density of the *densest subgraph* (Nash-Williams, 1961). It is widely known that real-world datasets such as social networks, road networks or planar graphs have very small arboricities (Eppstein et al., 2013).

*WL test.* The Weisfeiler–Leman test (WL test) is a family of combinatorial algorithms for distinguishing graphs through iterative refinement of vertex- or tuple-colorings (Weisfeiler & Leman, 1968; Cai et al., 1992). The most widely used variant is the 1WL test, or *color refinement*. Given a graph $G = (V, E)$, this algorithm assigns each vertex $v \in V$ a color $\mathsf{cr}^{(\ell)}(G, v)$ at iteration $\ell \geq 0$, defined inductively as follows: the initial color is constant, $\mathsf{cr}^{(0)}(G, v) := 1$, and the update rule is

$$\mathsf{cr}^{(\ell+1)}(G, v) := \big( \mathsf{cr}^{(\ell)}(G, v), \{\!\{ \mathsf{cr}^{(\ell)}(G, u) \mid u \in N(v) \}\!\} \big).$$

At each iteration $\ell$, the coloring induces a partition of the vertex set, where the partition at iteration $\ell + 1$ refines that at iteration $\ell$. Once this process stabilizes, we write $\mathsf{cr}(G, v)$ for the final color assigned to vertex $v$. We define $\mathsf{cr}(G)$ as the multiset $\{\!\{ \mathsf{cr}(G, v) \mid v \in V \}\!\}$, and call two graphs $G$ and $G'$ *distinguishable by 1WL* if $\mathsf{cr}(G) \neq \mathsf{cr}(G')$.

Due to lack of space, we present further preliminaries including the definition of chordal graphs, the $k$WL test and the NC-WL1 test in Appendix A. Additionally, we present all of our proofs in Appendix C.

## 3. Edge-based WL test

In this section, we introduce the EB-1WL test (for edge-based 1WL), a color refinement test that is more expressive than 1WL and NC-1WL and that colors edges rather than vertices — this is analogous to higher-order WL tests that

color vertex tuples. Unlike those tests, EB-1WL colors only edge pairs, reducing space complexity from quadratic in nodes to linear in edges, making it more practical.

Let $G = (V, E)$ be an undirected graph. We iteratively associate a color $\mathsf{eb}^{(\ell)}(G, (u, v))$ with each ordered pair $(u, v)$ such that $\{u, v\} \in E$. That is, each edge receives two colors, one for each ordering of its endpoints. The coloring of the ordered pair $(u, v)$ is defined inductively. Initially, every pair has the same color: $\mathsf{eb}^{(0)}(G, (u, v)) = 1$. At iteration $\ell + 1$, the color of $(u, v)$ is updated according to

$$\mathsf{eb}^{(\ell+1)}(G, (u, v)) = \Big( \mathsf{eb}^{(\ell)}(G, (u, v)), \tag{1}$$

$$\{\!\{ \mathsf{eb}^{(\ell)}(G, (u, x)) \mid x \in N(u) \}\!\}, \tag{2}$$

$$\{\!\{ \big( \mathsf{eb}^{(\ell)}(G, (u, y)), \mathsf{eb}^{(\ell)}(G, (v, y)) \big) \mid y \in N(v) \cap N(u) \}\!\}, \tag{3}$$

$$\{\!\{ \mathsf{eb}^{(\ell)}(G, (v, z)) \mid z \in N(v) \}\!\} \Big). \tag{4}$$

This update rule refines the color of an edge based on the colors of edges incident to its endpoints as can be seen in Fig. 1. Eq. (2) captures the influence of edges incident to $u$, Eq. (4) does the same for edges incident to $v$, and Eq. (3) encodes the interaction between edges that are incident to both $u$ and $v$, thereby forming a triangle and incorporating the local neighborhood structure around the edge.

We let $\mathsf{eb}(G, (u, v))$ denote the color of the ordered pair $(u, v)$ once the coloring partition of the edges defined by the EB-1WL test becomes stable. We write $\mathsf{eb}(G)$ for the multiset $\{\!\{ \mathsf{eb}(G, (u, v)), \mathsf{eb}(G, (v, u)) \mid \{u, v\} \in E \}\!\}$ and call two graphs $G$ and $G'$ *distinguishable by EB-1WL* if they have a different number of vertices or $\mathsf{eb}(G) \neq \mathsf{eb}(G')$.

**Computational cost of EB-1WL.** Next, we study the computational cost of EB-1WL. We consider an idealized computational model where a multiset of $s$ elements can be created in time $O(s)$ and stored in $O(1)$ space (in practice, this can be achieved with high probability using hashing). In this model, we need $O(m)$ space to store the graph and the colors of the edges.

Now we analyze the time needed to perform a single iteration. First, computing the multisets in Eq.s (2) and (4) for all nodes $u$ and $v$ can be done in total time $O(m)$, since for every node $u$ (and, resp., $v$) we spend time proportional to its degree. The more interesting part is computing Eq. (3) efficiently for all *ordered* edges $(u, v)$. For this, we need to go through all common neighbors $y$ of $u$ and $v$. A naïve approach would iterate over all neighbors $y$ of $u$ and then check whether $(y, v)$ exists. This becomes slow if $u$ has a much higher degree than $v$, potentially taking total time $O(md)$, where $d$ is the maximum degree of the graph. Instead, following the triangle-enumeration algorithm of Chiba & Nishizeki (1985), we iterate only over

the neighbors of the lower-degree endpoint of $(u, v)$ and check adjacency to the other endpoint. This still visits every common neighbor $y$ of $u$ and $v$ exactly once, so every triangle $(u, v, y)$ is included in the summation in Eq. (3); only the order and implementation of the enumeration change, and the permutation-invariant aggregation therefore remains unaffected. The analysis of Chiba & Nishizeki (1985) then implies a running time of $O(\alpha m)$, where $\alpha$ is the arboricity of the graph. Appendix B provides more details on this algorithm and its connection to EB-1WL.

**Proposition 1.** *An iteration of EB-1WL can be performed using $O(m)$ space and $O(\alpha m)$ time.*

We note that this running time is highly efficient in practice and that summing over all triangles polynomially faster than in $O(\alpha m)$ time would violate established conjectures from the computational complexity community (Kopelowitz et al., 2016; Vassilevska Williams & Xu, 2020).

## 4. The distinguishing power of EB-1WL

We present our theoretical results on the expressiveness of EB-1WL, ranging from a comparison with 1WL and NC-1WL (Sec. 4.1), over a logical characterization (Sec. 4.2) to its power based on homomorphism counts (Sec. 4.3).

### 4.1. Expressivity

We start by showing that the EB-1WL test is strictly more expressive than 1WL and the edge-based NC-1WL test by Liu et al. (2024) in distinguishing non-isomorphic graphs.

**Theorem 2.** *Every pair of graphs distinguishable by NC-1WL is also distinguishable by EB-1WL. Additionally, the graphs $G$ and $H$ in Fig. 2 can be distinguished by EB-1WL but not by NC-1WL. Thus, EB-1WL is strictly more expressive than NC-1WL and 1WL.*

Interestingly, in Appendix F we also show that EB-1WL is strictly more expressive than 1WL with triangle counts added on node and edge level. Furthermore, we note that any pair of graphs distinguishable by EB-1WL is also distinguishable by 2WL, though the reverse implication does not hold. A concrete example of a pair of graphs that is distinguishable by 2WL but neither by EB-1WL nor NC-1WL is shown in Fig. 3.

### 4.2. Logical characterization

Next, we provide a complete characterization of EB-1WL's expressiveness through first-order logic. To highlight the merit of this result, we note that the distinguishing power of the $k$-WL test can be characterized via a fragment of first-order logic, namely the $(k+1)$-*variable fragment with counting quantifiers* (Cai et al., 1992), with 1WL and 2WL corresponding to the two- and three-variable fragments, re-

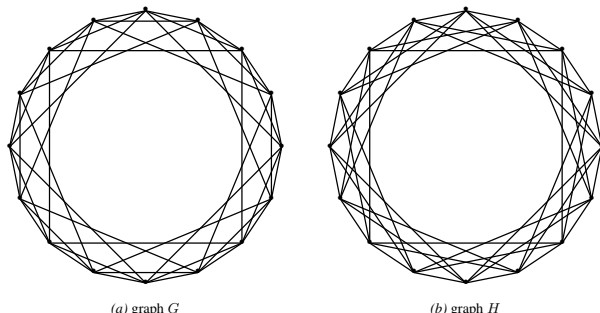

*(a)* graph $G$          *(b)* graph $H$

*Figure 2.* The graphs $G$ and $H$ from the proof of Theorem 2.

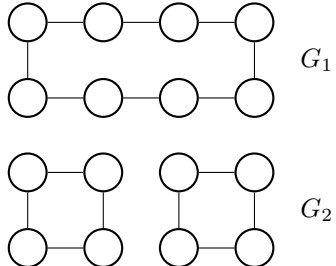

*Figure 3.* Graphs distinguishable by 2WL but not by EB-1WL.

spectively. These logical characterizations have been an important tool in studying GNNs, as they offer insights into what GNNs can accomplish (cf., (Barceló et al., 2020)). We show that EB-1WL also admits a natural logical characterization, further demonstrating the robustness and naturalness of our newly proposed framework. Specifically, EB-1WL corresponds to a novel natural logic we introduce below. This logic lies strictly between the two- and three-variable counting fragments, providing a precise characterization of its position between 1WL and 2WL.

In the following, we assume familiarity with the syntax and semantics of first-order logic (FO). We let $\phi$ be a formula and write $\phi(\bar{x})$ to indicate that the free variables of $\phi$ are exactly those in the tuple $\bar{x}$. When interpreted over graphs, FO formulas are defined over a vocabulary consisting of a single binary relation symbol $E$. Specifically, the formula $E(x, y)$ is interpreted over a graph $G = (V, E)$ as the set of all pairs $(u, v) \in V \times V$ such that $\{u, v\} \in E$. Given a tuple $\bar{x} = (x_1, \ldots, x_n)$ of distinct variables, we define the FO formula

$$\mathsf{clique}(\bar{x}) := \bigwedge_{1 \leq i < j \leq n} E(x_i, x_j),$$

so that its interpretation over $G$ is the set of all $n$-tuples of vertices that form a clique in $G$.

We now introduce our novel *clique-based first-order logic with counting (CFOC)* via the following syntax:

1. If $\bar{x}$ is a tuple of distinct variables, then $\mathsf{clique}(\bar{x})$ is a formula in CFOC.

2. If $\phi(\bar{x})$ is a formula in CFOC, then so is $\mathrm{clique}(\bar{x}) \wedge \neg\phi(\bar{x})$.

3. If $\phi(\bar{x})$ and $\psi(\bar{y})$ are formulas in CFOC, then so is $\mathrm{clique}(\bar{z}) \wedge (\phi(\bar{x}) \star \psi(\bar{y}))$ for any $\star \in \{\wedge, \vee\}$, where $\bar{z}$ is the tuple obtained by collecting all variables that occur in $\bar{x}$ and $\bar{y}$, removing any duplicates, so that each variable appears exactly once.

4. If $\phi(\bar{x}, y)$ is a formula in CFOC, then $\mathrm{clique}(\bar{x}) \wedge \exists^{\geq k} y \, \phi(\bar{x}, y)$ is a formula in CFOC, for any integer $k \geq 1$, where the semantics of $\exists^{\geq k} y \, \phi(\bar{x}, y)$ assert that there exist at least $k$ vertices $v$ such that $\phi(\bar{x}, y)$ holds when $y = v$.

Intuitively, CFOC restricts FO with counting quantifiers to formulas whose free variables form a clique in the graph. We note that this is reminiscent of, but not directly related to, the so-called *clique-guarded fragments* of first-order logic; see, e.g., Grädel (1999).

We denote $\mathrm{CFOC}^3$ the fragment of CFOC that consists of formulas that use at most three variables. For example, the $\mathrm{CFOC}^3$ formula $\psi := \exists^{\geq 1} x \exists^{\geq 1} y \, \big( \mathrm{clique}(x, y) \wedge \exists^{\geq 3} z \, \mathrm{clique}(x, y, z) \big)$ checks if there is an edge which is a part of at least 3 triangles. A $\mathrm{CFOC}^3$ *sentence* is a $\mathrm{CFOC}^3$ formula without free variables. We call graphs $G$ and $H$ *distinguishable by* $\mathrm{CFOC}^3$, if there is a $\mathrm{CFOC}^3$ sentence $\phi$ such that $G \models \phi$ but $H \not\models \phi$. We can now establish our characterization of EB-1WL.

**Theorem 3.** *The pairs of graphs that are distinguishable by EB-1WL are precisely those that are distinguishable by* $\mathrm{CFOC}^3$.

*Proof sketch.* To prove the theorem, we show for every pair of graphs $G, G'$ that $\mathrm{eb}(G) = \mathrm{eb}(G')$ if and only if $G \models \phi \Leftrightarrow G' \models \phi$ for every $\mathrm{CFOC}^3$ sentence. We use two lemmas to prove this theorem which both make use of connections between $t \geq 0$ rounds of EB-1WL and $t$-types.

($\Leftarrow$). Lemma 9 states that for every individual color computed in $t$ rounds of EB-1WL, there exists a corresponding $\mathrm{CFOC}^3$ formula of quantifier depth $t$. To prove this direction, we assume that every $\mathrm{CFOC}^3$ sentence has identical truth values on both graphs. Recall that $\mathrm{eb}(G)$ is the multiset of all colors of edges EB-1WL has computed after stabilizing at some iteration $t$. For every color in $\mathrm{eb}(G)$ we can use the lemma to construct a sentence of quantifier depth $t$ that corresponds to this color and the multiplicity of that color in $\mathrm{eb}(G)$. By combining these sentences, we obtain a $\mathrm{CFOC}^3$ sentence $\phi_G$ which encodes the multiset of colors of EB-1WL on $G$ and has quantifier depth $t + 2$, accounting for the two outer counting quantifiers needed to bind the edge variables $(x, y)$ over the pooling step. Since $\phi_G$ determines the colors and multiplicities of EB-1WL by construction, it follows that $G \models \phi_G$. By assumption both graphs agree on all sentences which means that $G \models \phi_G$ im-

plies $G' \models \phi_G$. As a consequence it follows that EB-1WL has the same colors on both $G$ and $G'$.

($\Rightarrow$). Lemma 10 states that if $t$-rounds of EB-1WL cannot distinguish a graph pair, then both graphs have the same $t$-type. Let $\phi$ be an arbitrary $\mathrm{CFOC}^3$ sentence of quantifier depth $t$. By assumption we know that $\mathrm{eb}(G) = \mathrm{eb}(G')$ and thus that for every $t' \geq 0$ it holds that $\mathrm{eb}^{t'}(G) = \mathrm{eb}^{t'}(G')$. Hence, from the lemma it follows that $G \models \phi \Leftrightarrow G' \models \phi$ which finishes the proof. $\square$

**Example 1.** The sentence $\psi$ shown above holds in graph $H$ from Figure 2, but not in graph $G$. Thus, EB-1WL is able to distinguish these graphs. $\blacktriangle$

### 4.3. Distinguishing power based on homomorphism counts

Next, we study the expressiveness of EB-1WL through the lens of homomorphism counts, which has recently become an important theme in the study of the WL test (cf., Dell et al. (2018); Barceló et al. (2021); Jin et al. (2024); Bao et al. (2025)).

Formally, a *homomorphism* from a graph $G = (V, E)$ to a graph $H = (V', E')$ is a mapping $h : V \to V'$ such that $\{h(u), h(v)\} \in E'$ for every edge $\{u, v\} \in E$. The connection between WL and homomorphism counts is as follows: two graphs are distinguishable by $k$WL if and only if they differ in the number of homomorphisms from some graph of *treewidth* at most $k$ (Dvořák, 2010; Dell et al., 2018). For $k = 1$, this corresponds to the class of trees, and for $k = 2$ to the class of series-parallel graphs.

We show that distinguishability by EB-1WL is at least as powerful as distinguishing graphs by homomorphism counts from the class of *chordal* graphs of treewidth two, which strictly lies between the classes of graphs of treewidth one and two. This offers additional support for viewing EB-1WL as a natural and well-founded counterpart to the standard 1WL test.

**Theorem 4.** *If two graphs have a different number of homomorphisms from some chordal graph of treewidth at most 2, they are distinguishable by EB-1WL.*

## 5. Edge-based Graph Neural Networks

Now we introduce the *EB-GNN architecture*, a message-passing framework whose expressive power coincides with that of the EB-1WL test.

Formally, a $d$-dimensional *EB-GNN* $\mathcal{T}$ with $L > 0$ layers is specified by parameters $a_i, b_i, c_i, u_i, v_i \in \mathbb{R}^d$, $A_i, C_i, U_i, V_i \in \mathbb{R}^{d \times d}$ and $B_i \in \mathbb{R}^{d \times 2d}$ for $i = 1, \ldots, L$. Given a graph $G = (V, E)$, the EB-GNN $\mathcal{T}$ assigns to each ordered edge $(u, v)$ with $\{u, v\} \in E$ and each layer

$0 \leq i \leq L$ a feature vector $f^{(i)}(u,v) \in \mathbb{R}^d$.

At the input layer, we set[4]

$$f^{(0)}(u,v) = \begin{pmatrix} 1 & 0 & \dots & 0 \end{pmatrix}^T \in \mathbb{R}^d. \quad (5)$$

For $1 \leq i \leq L$, the update rules are given by

$$\alpha^{(i)}(u) = \sum_{x \in N(u)} \mathrm{ReLU}\Big(A_i \cdot f^{(i-1)}(u,x) + a_i\Big), \quad (6)$$

$$\beta^{(i)}(u,v) = \sum_{y \in N(u) \cap N(v)} \mathrm{ReLU}\Big(B_i \cdot \begin{pmatrix} f^{(i-1)}(u,y) \\ f^{(i-1)}(v,y) \end{pmatrix} + b_i\Big), \quad (7)$$

$$\gamma^{(i)}(v) = \sum_{z \in N(v)} \mathrm{ReLU}\Big(C_i \cdot f^{(i-1)}(v,z) + c_i\Big), \quad (8)$$

$$g^{(i)}(u,v) = f^{(i-1)}(u,v) + \alpha^{(i)}(u) + \beta^{(i)}(u,v) + \gamma^{(i)}(v), \quad (9)$$

$$f^{(i)}(u,v) = g^{(i)}(u,v) + \mathrm{FFN}_{U_i,u_i,V_i,v_i}(g^{(i)}(u,v)), \quad (10)$$

where $\mathrm{ReLU}(x) = \max\{0,x\}$ is applied coordinate-wise, and $\mathrm{FFN}_{U,u,V,v}(x) = V \cdot \mathrm{ReLU}(Ux+u) + v$.

The overall output of $\mathcal{T}$ on $G$ is defined as

$$\mathcal{T}(G) = \sum_{\{u,v\} \in E} f^{(L)}(u,v). \quad (11)$$

We say that graphs $G$ and $H$ are *distinguishable by EB-GNNs* if there exists an EB-GNN $\mathcal{T}$ with $\mathcal{T}(G) \neq \mathcal{T}(H)$.

It is immediate that the distinguishing power of EB-GNNs cannot exceed that of EB-1WL. Notably, we can show that this upper bound is tight.

**Theorem 5.** *Pairs of graphs distinguishable by EB-1WL are also distinguishable by EB-GNNs.*

**Algorithmic implementation.** In our implementation we perform a preprocessing step in which we enumerate all triangles in time $O(\alpha m)$ using the algorithm of Chiba & Nishizeki (1985). Since each triangle $(u,v,y)$ corresponds to an edge $(u,v)$ and a node $y \in N(u) \cap N(v)$, we can obtain the sets $N(u) \cap N(v)$ required in Eq. (7) for all edges $(u,v)$ in time $O(\alpha m)$. After that, each iteration of EB-GNN takes time $O(m+t)$, where $t$ is the number of triangles—implying that in practice we might obtain better running times per iteration than in Proposition (1) because $t = O(\alpha m)$. However, due to this preprocessing step we require memory $O(m+t)$ (instead of just $O(m)$).

*Table 1.* Empirical results on expressivity datasets. GIN + $C_3$ is an MPNN that uses triangle subgraph counts as additional node features. MPNN and 2WL results on BREC are from Wang & Zhang (2024). For details on runtime constants see Table 4.

| Model | Runtime | CSL Accuracy (↑) | BREC # Distinguishable Graph Pairs (↑) | | | |
|---|---|---|---|---|---|---|
| | | | Basic | Reg. | Ext. | CFI |
| MPNN | $\mathcal{O}(m)$ | 10% | 0 | 0 | 0 | 0 |
| MPNN + $C_3$ | $\mathcal{O}(\alpha m)$ | 20% | 0 | 0 | 0 | 0 |
| 2WL | $\mathcal{O}(n^3)$ | – | 60 | 50 | 100 | 60 |
| $I^2$-GNN | $\mathcal{O}(nsd^2)$ | – | 60 | 100 | 100 | 21 |
| DRFWL | $\mathcal{O}(nd^4)$ | – | 60 | 50 | 99 | 0 |
| 4-$\ell$-GIN | $\mathcal{O}(nd^3)$ | 60% | 60 | 100 | 95 | 2 |
| NC-GNN | $\mathcal{O}(\alpha m)$ | 20% | 52 | 48 | 0 | 0 |
| EB-GNN (ours) | $\mathcal{O}(\alpha m)$ | 20% | 59 | 48 | 60 | 0 |

## 6. Experimental Evaluation

Now we empirically evaluate EB-GNN. The primary goal of our experiments is to show that EB-GNN provides a fast and expressive general-purpose GNN architecture. Therefore, we evaluate EB-GNN across tasks from diverse domains. We compare EB-GNN against Message Passing Neural Networks (MPNNs), another widely used general-purpose architecture, as well as against state-of-the-art models specifically optimized for each corresponding task. Our implementation is available at https://github.com/ocatias/EdgeBasedGNNs.

We focus on graph-level predictions and predictions on existing edges. While graph-level prediction tasks are a well-established use case for predictive GNNs (Morris et al., 2019; Paolino et al., 2024; Southern et al., 2025), prediction tasks on existing edges have received less attention. This latter task is relevant in chemistry: rather than relying on costly molecular simulations, GNNs can directly predict quantum mechanical properties, e.g., the bond length between atoms (Li et al., 2024).

**Experiment setup.** Next, we describe the datasets and baseline models used for comparison. We evaluate EB-GNN on two synthetic datasets designed to measure its practically realized expressivity. Additionally, we assess performance on three real-world datasets: two composed of small molecular graphs and one consisting of large cybersecurity graphs. Consistent with previous observations that real-world graphs have low arboricity, we find that these datasets exhibit small arboricity: $\leq 3$ for molecular graphs and $\leq 15$ for large cybersecurity graphs with each dataset having $< 4$ mean arboricity. For each dataset, we compare EB-GNN against both general-purpose models and state-of-the-art task-specific baselines. Details on model and experiment setup are in App. D

---

[4]When nodes or edges have additional input features, one can take them into account while defining $f^{(0)}(u,v)$, see Section 6.

*Table 2.* Results on `MalNet-Tiny`. Top three models as 1st, 2nd, 3rd. Baselines from Southern et al. (2025).

| Method | MalNet-Tiny $_{Accuracy\ (\uparrow)}$ |
|--------|-----------------------|
| MPNN | 91.10 $_{\pm 0.98}$ |
| HyMN | 92.84 $_{\pm 0.52}$ |
| GPS (Perf.) | 92.14 $_{\pm 0.24}$ |
| GPS (BigBird) | 91.02 $_{\pm 0.48}$ |
| GPS (Transf.) | 90.85 $_{\pm 0.68}$ |
| NC-GNN | 92.50 $_{\pm 0.56}$ |
| EB-GNN | 93.30 $_{\pm 0.66}$ |

*Synthetic datasets for measuring expressivity.* We evaluate the empirical expressivity of EB-GNN on two synthetic datasets: `CSL` (Murphy et al., 2019; Dwivedi et al., 2023) and `BREC` (Wang & Zhang, 2024). The `CSL` dataset consists of 150 graphs with 41 nodes each, grouped into 10 distinct isomorphism classes. These classes, defined by skip connections between Hamiltonian cycles, are all indistinguishable by 1WL. The graph-level task is to classify each graph according to its isomorphism class. The `BREC` dataset comprises pairs of graphs that are indistinguishable by 1WL but distinguishable by 3WL. For each pair, the graph-level task is to compute embeddings that correctly differentiate the two graphs.

On the synthetic datasets, we compare EB-GNN against 2WL to assess how much of 2WL's expressivity EB-GNN can replicate while maintaining asymptotically faster runtimes. In addition, we include a comparison with the MPNN GIN (Xu et al., 2019) augmented with triangle subgraph counts as node features (Bouritsas et al., 2022), referred to as MPNN + $C_3$. This comparison helps isolate how much of EB-GNN's expressivity gain stems from its ability to count triangles, as captured by the $\beta$ aggregation in Eq. (7). We also compare against NC-GNN to determine whether our theoretical increase in expressivity (Thm. 2) can also be measured empirically. Furthermore, we compare against all models used as baselines in other experiments for which we could find publicly available results.

*Molecular edge-level and graph-level tasks.* We further evaluate EB-GNN on a range of molecular edge-level and graph-level prediction tasks. Li et al. (2024) introduce the `QMD` dataset, which contains 65 000 molecular graphs annotated with various quantum mechanical properties. We assess EB-GNN on all four edge-level regression tasks from `QMD`. We evaluate four diverse graph-level regression tasks from `QMD`, selected to represent varied objectives (most other graph-level tasks in `QMD` focus on predicting HOMO/LUMO gaps). We also evaluate EB-GNN on 12 graph-level regression tasks from the widely used `QM9` dataset (Wu et al., 2018), following common practice (Morris et al., 2019; Paolino

et al., 2024). `QM9` comprises 130 000 molecular graphs representing molecules of up to 9 atoms. On average, graphs in both `QMD` and `QM9` contain fewer than 20 nodes.

For `QMD` tasks, we compare EB-GNN against D-MPNN (Yang et al., 2019; Dai et al., 2016), a directed MPNN that has demonstrated strong performance on chemical prediction tasks (Vermeire et al., 2022; Heid & Green, 2022) and is integrated into the widely adopted ChemProp framework (Heid et al., 2024). Similar to EB-GNN, D-MPNN performs message passing on edges rather than nodes. Furthermore, we also compare against NC-GNN (Liu et al., 2024) which we train using the same hyperparameter tuning procedure as EB-GNN. For `QM9`, we compare against a standard MPNN and several expressive GNN architectures: 1-2-3 GNN (Morris et al., 2019), DTNN (Schütt et al., 2017; Wu et al., 2018), NestedGNN (Zhang & Li, 2021), I2-GNN (Huang et al., 2023), DRFWL (Zhou et al., 2023), and the recent state-of-the-art 5-$\ell$GIN baseline (Paolino et al., 2024).

*Cybersecurity.* To evaluate our model on a different domain with larger graphs, we conduct experiments on `MalNet-Tiny` (Freitas et al., 2021). `MalNet-Tiny` consists of 5 000 graphs with an average of over 1 500 nodes, sampled from the `MalNet` dataset. The graph-level task is to classify whether a function call is benign or belongs to one of four malicious classes (AdWare, Trojan, Addisplay, Downloader). For comparison, we include a standard MPNN and the recently proposed subgraph GNN HyMN (Southern et al., 2025). We further compare against the graph transformer GPS (Rampášek et al., 2022) using different attention mechanisms: Performer (Choromanski et al., 2021), Big Bird (Zaheer et al., 2020), and the standard Transformer (Vaswani et al., 2017). Finally, we also compare against NC-GNN (Liu et al., 2024) which we train in the same fashion as EB-GNN.

**Results.** Recall that comparisons are made against both general-purpose MPNNs and state-of-the-art models for each dataset. Our experiments demonstrate that EB-GNN consistently outperforms standard MPNNs and remains competitive with state-of-the-art approaches. We also evaluate the speed of our method in App. E.

*Results on synthetic data.* Table 1 presents the results of our expressivity experiments on the synthetic datasets. On `CSL`, EB-GNN achieves an accuracy of 20%, outperforming a vanilla MPNN (10% accuracy) and matching an MPNN augmented with triangle counts (MPNN + $C_3$, 20% accuracy). On `BREC`, the gains of EB-GNN cannot be attributed to triangle counting, as evidenced by the MPNN + $C_3$ results. Remarkably, EB-GNN achieves performance on `Basic` and `Regular` graphs that is nearly identical to 2WL, though it fails to distinguish any `CFI` pairs. These results indicate that EB-GNN's empirically realized expressivity extends well beyond triangle counting and, in many cases, approaches

*Table 3.* MAE ($\downarrow$) on QMD. Best model per task marked blue. D-MPNN results from Li et al. (2024).

| Model | Edge-level Tasks (MAE $\downarrow$) | | | | Graph-level Tasks (MAE $\downarrow$) | | | |
|---|---|---|---|---|---|---|---|---|
| | Bond Index (unitless) $\times 10^{-3}$ | Bond Length (Å) $\times 10^{-3}$ | Bonding Electrons (e) $\times 10^{-2}$ | Natural Ionicity (unitless) $\times 10^{-4}$ | IP $\times 10^{-3}$ | EA $\times 10^{-3}$ | Dipole Moment (debye) $\times 10^{-1}$ | Traceless Quad. Mom. (debye Å) $\times 10^{0}$ |
| D-MPNN | 6.65 | 4.48 | 1.46 | 9.00 | 4.29 | 4.06 | 4.59 | 1.62 |
| NC-GNN | 4.82 $\pm 0.04$ | 3.33 $\pm 0.03$ | 1.28 $\pm 0.01$ | 5.39 $\pm 0.25$ | 5.3 $\pm 0.03$ | 4.35 $\pm 0.03$ | 4.45 $\pm 0.02$ | 1.57 $\pm 0.01$ |
| EB-GNN | 4.42 $\pm 0.01$ | 3.17 $\pm 0.011$ | 1.19 $\pm 0.018$ | 4.64 $\pm 0.01$ | 5.49 $\pm 0.26$ | 4.46 $\pm 0.12$ | 4.33 $\pm 0.05$ | 1.55 $\pm 0.01$ |

2WL expressivity while maintaining significantly faster runtimes. We outperform NC-GNN in two categories and tie it in the remaining three. Notably, on Extension graphs EB-GNN solves 60 instances while NC-GNN solves 0, highlighting our empirical increase in expressivity.

*Molecular edge-level and graph-level tasks.* Table 3 presents the edge-level and graph-level results on QMD, compared against the baseline D-MPNN (Li et al., 2024). EB-GNN outperforms D-MPNN across all edge-level tasks, reducing the mean absolute error by 20% to 50% depending on the task. On graph-level tasks, EB-GNN and D-MPNN perform comparably, with EB-GNN slightly outperforming D-MPNN on two tasks and slightly underperforming on the other two. EB-GNN outperforms NC-GNN in six out of eight datasets and only loses to NC-GNN for datasets where D-MPNN is the best performing model. Table 4 summarizes the results on QM9, where we compare against several expressive GNNs. The current state-of-the-art on this dataset is 5-$\ell$GNN (Paolino et al., 2024), which significantly improved over previous architectures. EB-GNN ranks as the best or second-best model on 11 out of 12 tasks, achieving comparable or better performance than 5-$\ell$GNN on 7 tasks (with at most 6% lower accuracy) and outperforming it on 4 tasks. Moreover, EB-GNN is substantially more computationally efficient than 5-$\ell$GNN: while 5-$\ell$GNN has asymptotic runtime $\mathcal{O}\left(nd^5\right)$, EB-GNN only requires $\mathcal{O}\left(\alpha m\right)$ time. In our experiments, EB-GNN is approximately 4 times faster (see App. E).

*Results on graph-level tasks for malware detection.* Table 2 shows the results on MalNet-Tiny. EB-GNN outperforms all other models, including a standard MPNN, various graph transformers (GPS), and the subgraph GNN HyMN, demonstrating that EB-GNN generalizes well to other domains and scales effectively to larger graphs.

## 7. Conclusion, Limitations and Future work

We propose an edge-based message passing algorithm that combines high expressivity with near-linear runtime on sparse graphs. Our analysis fully characterizes its expressiv-

ity in terms of first-order logic and establishes a lower bound via homomorphism counting. Empirically, our architecture outperforms standard MPNNs while remaining competitive with more expressive models, at a substantially lower runtime. We observe that EB-GNN is a general-purpose architecture: while it achieves strong empirical results, it does not rely on specialized techniques used on top of basic architectures known to improve GNN performance, such as using subgraph counts (Bouritsas et al., 2022), homomorphism counts (Barceló et al., 2021; Welke et al., 2023; Jin et al., 2024) or positional encodings (You et al., 2019; Ying et al., 2021; Ma et al., 2023; Bao et al., 2025).

*Limitations.* Although EB-1WL and EB-GNN achieve near-linear running times for sparse graphs, their efficiency depends on arboricity and triangle enumeration, which can be expensive on very dense graphs, potentially limiting scalability in such settings. Furthermore, EB-GNN produces embeddings only for edges preventing direct application to node-level tasks and standard link-prediction methods that rely on node embeddings. We leave extending EB-GNN to these tasks as future work.

*Future work.* An interesting open problem left by our work is whether the converse of Thm. 4 holds, i.e., whether graphs distinguishable by EB-1WL are exactly those distinguishable by homomorphism counts from some chordal graph of treewidth 2. Another line of future work concerns the tradeoff between expressiveness and generalization: recent results show that greater expressiveness need not harm generalization if matched to task demands and training data (Maskey et al., 2026), and can even help when graphs are well separated by large margins (Li et al., 2025). EB-GNNs strike a principled balance — more expressive than NC-1WL, yet far cheaper than 2WL — while showing strong results across benchmarks. Future work should broaden empirical evaluation to fully assess this balance of expressiveness, scalability, and generalization.

*Table 4.* Normalized test MAE ($\downarrow$) on QM9 dataset. Top three models as 1st , 2nd , 3rd . Table based on Paolino et al. (2024). For runtime, $n$ is the number of nodes, $m$ the number of edges, $c$ and $s$ are maximum size of subgraph sizes, $d$ the maximum degree, and $\alpha$ the arboricity.

| Target (MAE $\downarrow$) | Model | | | | | | | |
|---|---|---|---|---|---|---|---|---|
| Runtime | MPNN $\mathcal{O}(m)$ | 1-2-3 GNN $\mathcal{O}(n^3)$ | DTNN $\mathcal{O}(m)$ | NestedGNN $\mathcal{O}(ncd)$ | I2-GNN $\mathcal{O}(nsd^2)$ | DRFWL $\mathcal{O}(nd^4)$ | 5-$\ell$GIN $\mathcal{O}(nd^5)$ | EB-GNN $\mathcal{O}(\alpha m)$ |
| $\mu\ (\times 10^{-1})$ | 4.93 | 4.76 | 2.44 | 4.28 | 4.28 | 3.46 | 3.50 $\pm 0.11$ | 3.15 $\pm 0.02$ |
| $\alpha\ (\times 10^{-1})$ | 7.8 | 2.7 | 9.5 | 2.90 | 2.30 | 2.22 | 2.17 $\pm 0.25$ | 2.08 $\pm 0.03$ |
| $\varepsilon_{\text{homo}}\ (\times 10^{-3})$ | 3.21 | 3.37 | 3.88 | 2.65 | 2.61 | 2.26 | 2.05 $\pm 0.05$ | 2.18 $\pm 0.01$ |
| $\varepsilon_{\text{lumo}}\ (\times 10^{-3})$ | 3.55 | 3.51 | 5.12 | 2.97 | 2.67 | 2.25 | 2.16 $\pm 0.04$ | 2.17 $\pm 0.02$ |
| $\Delta(\varepsilon)\ (\times 10^{-3})$ | 4.9 | 4.8 | 11.2 | 3.8 | 3.8 | 3.24 | 3.21 $\pm 0.14$ | 3.02 $\pm 0.01$ |
| $R^2$ | 34.1 | 22.9 | 17.0 | 20.5 | 18.64 | 15.04 | 13.21 $\pm 0.19$ | 13.83 $\pm 0.12$ |
| ZVPE $(\times 10^{-4})$ | 12.4 | 1.9 | 17.2 | 2. | 1.4 | 1.7 | 1.27 $\pm 0.03$ | 1.26 $\pm 0.01$ |
| $U_0$ | 2.32 | 0.0427 | 2.43 | 0.295 | 0.211 | 0.156 | 0.0418 $\pm 0.0520$ | 0.063 $\pm 0.002$ |
| $U$ | 2.08 | 0.111 | 2.43 | 0.361 | 0.206 | 0.153 | 0.023 $\pm 0.023$ | 0.078 $\pm 0.008$ |
| $H$ | 2.23 | 0.0419 | 2.43 | 0.305 | 0.269 | 0.145 | 0.0352 $\pm 0.0304$ | 0.0564 $\pm 0.0075$ |
| $G$ | 1.94 | 0.0469 | 2.43 | 0.489 | 0.261 | 0.156 | 0.0118 $\pm 0.0015$ | 0.076 $\pm 0.006$ |
| $C_v$ | 0.27 | 0.0944 | 2.43 | 0.174 | 0.0730 | 0.0901 | 0.0702 $\pm 0.0024$ | 0.092 $\pm 0.001$ |

# Acknowledgements

We are grateful to Esther Heid for helpful discussions on the role of edge-based GNNs in chemistry.

This research has been funded by the Vienna Science and Technology Fund (WWTF) [Grant ID: 10.47379/VRG23013] and [Grant ID: 10.47379/ICT2201]. Kozachinskiy is supported by ANID Fondecyt Iniciación grant 11250060. Barceló, Kozachinskiy, and Rojas are funded by the National Center for Artificial Intelligence CENIA FB210017, Basal ANID. Barceló is also funded by ANID Millennium Science Initiative Program Code ICN17002.

# Impact Statement

This paper presents work whose goal is to advance the field of Machine Learning. There are many potential societal consequences of our work, none of which we feel must be specifically highlighted here.

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

**Use of LLMs in paper writing.** We used LLMs to polish sentences and improve phrasing.

## A. Further Preliminaries

A graph is *chordal* if it has no induced subgraph that is a cycle of length 4 or larger. Note that, by definition, trees are always chordal.

*The kWL test.* Higher-order versions of the WL test are also widely studied in the literature. Rather than focusing on individual vertices, the $k$WL test, for $k > 1$, considers $k$-tuples of vertices. Each $k$-tuple is assigned a color that reflects the isomorphism type of the subgraph induced by those vertices. At each refinement step, the color of a $k$-tuple is updated by considering all possible ways of replacing one of its vertices with another vertex in the graph. For example, when $k = 2$, each ordered pair $(u, v)$ refines its color by looking at pairs such as $(u, w)$ and $(w, v)$ for all possible $w$. In this way, the $k$WL test captures not only the view of individual vertices, but also the structural relations within patterns of size $k$.

*Neighbor-communication WL test.* An extension of 1WL that incorporates information about the edges between the neighbors of a node has been proposed recently (Liu et al., 2024). The resulting test, called *NC-1WL test*, where NC stands for *neighbor communication*, proceeds similarly to 1WL, but updates the color $\mathsf{nc}^{(\ell)}(G, v)$ of each vertex $v$ in a graph $G = (V, E)$ according to the rule

$$\mathsf{nc}^{(\ell+1)}(G, v) := \big( \mathsf{nc}^{(\ell)}(G, v), \{\!\{\mathsf{nc}^{(\ell)}(G, u) \mid u \in N(v)\}\!\},$$
$$\{\!\{(\mathsf{nc}^{(\ell)}(G, u), \mathsf{nc}^{(\ell)}(G, w)) \mid u, w \in N(v), \{u, w\} \in E\}\!\}\big).$$

In other words, besides taking into account the multiset of colors of the neighbors of a vertex $v$, as in 1WL, the NC-1WL test also considers the multiset of color pairs corresponding to the edges within the neighborhood of $v$. We let $\mathsf{nc}(G, v)$ denote the color of node $v$ once the coloring partition on vertices defined by the NC-1WL test becomes stable. We write $\mathsf{nc}(G)$ for the multiset $\{\!\{\mathsf{nc}(G, v) \mid v \in V\}\!\}$, and call two graphs $G$ and $G'$ *distinguishable by NC-1WL* if $\mathsf{nc}(G) \neq \mathsf{nc}(G')$.

The additional structural information collected by NC-1WL makes it strictly more powerful than 1WL in distinguishing certain classes of non-isomorphic graphs. In turn, every pair of graphs that can be distinguished by NC-1WL can also be distinguished by 2WL, but the converse does not hold. The advantage of NC-1WL over 2WL, however, is that it achieves stronger discriminative power while still operating at the *vertex level*, in the same spirit as 1WL. In particular, its computational cost is closer to that of 1WL than to the more demanding 2WL procedure.

## B. Relationship with the Chiba–Nishizeki algorithm

We provide a thorough explanation of the algorithm by Chiba & Nishizeki (1985), state some of its properties which are important for our architecture and describe how the algorithm inspired our GNN architecture.

**Description of the Chiba–Nishizeki algorithm.** The algorithm by Chiba & Nishizeki (1985) obtains as input an undirected, unweighted graph $G = (V, E)$ and it returns a list of all triangles in $G$. Here, we provide a slightly modified version of the algorithm, which is slightly easier to describe but has the same properties.

Concretely, the algorithm works as follows. It iterates over all edges $(u, v) \in E$ and for each edge $(u, v)$ does the following: Suppose w.l.o.g. that $u$ is the lower-degree endpoint of the endpoint of the edge $(u, v)$, i.e., assume that $|N(u)| \leq |N(v)|$. Now the algorithm iterates over all neighbors $w \in N(u)$ and checks if $w \in N(v)$; if this is the case then it returns that $(u, v, w)$ is a triangle.

Note that the algorithm returns all triangles: Clearly, for any triangle $(u, v, w)$ its edge $(u, v)$ will be considered in the outer loop and we will have that $w \in N(u)$ (thus $w$ is considered in the inner loop) and $w \in N(v)$ (satisfying the if-condition). Thus, the triangle $(u, v, w)$ will be reported.

Further, the algorithm's running time is $O(\alpha m)$: Note that for each edge $(u, v)$ we spend time $O(\min\{|N(u)|, |N(v)|\})$ since we only spend time proportional to the neighborhood size of the lower-degree endpoint of $(u, v)$. Here, we use that the check whether $w \in N(v)$ can be done in time $O(1)$ using hash maps.[5] Thus, its total running time is given by

---

[5]This is where our version of the algorithm differs from the original. The paper by Chiba & Nishizeki (1985) does not use hash maps and instead uses a slightly more complicated marking procedure.

$\sum_{(u,v)\in E} O(\min\{|N(u)|, |N(v)|\})$ and Chiba & Nishizeki (1985, Lemma 2) showed that this quantity is bounded by $O(\alpha m)$.

**Properties of the Chiba–Nishizeki algorithm.** The algorithm has several important properties:

1. The algorithm lists all triangles in time $O(\alpha m)$. This implies that graphs with arboricity $\alpha$ contain at most $O(\alpha m)$ triangles. This allows us to bound the number of messages we need to send in our architecture due to triangles by $O(\alpha m)$.

2. The running time of the Chiba–Nishizeki algorithm is optimal under standard assumptions from the complexity theory community (Kopelowitz et al., 2016; Vassilevska Williams & Xu, 2020). Thus, the preprocessing time of our algorithm cannot be improved if all triangles need to be enumerated.

3. It is well-known in the algorithms community that the class of graphs with arboricity $O(1)$ contains natural graph families, such as planar graphs, minor-closed families, and preferential attachment graphs, among others. Thus, for all graphs from these families, all triangles can be enumerated in time $O(n)$ and thus in time linear in the size of the input graph. See, e.g., Chiba & Nishizeki (1985, Lemma 1) for the planar case.

4. It holds that $\alpha = O(\sqrt{m})$ Chiba & Nishizeki (1985, Lemma 1). As a consequence, the arboricity can be at most a $O(\sqrt{n})$ factor larger than the average degree in the graph. Indeed, this is the case for the simple example consisting of a path with $n - \sqrt{n}$ vertices, which is connected to a clique with $\sqrt{n}$ vertices; this graph has $m = O(n)$ edges and thus average degree $O(1)$ but its arboricity is $O(\sqrt{n})$ due to the clique.

5. The arboricity is highly related to other graph parameters, such as the *degeneracy* or the *densest subgraph* (Nash-Williams, 1961) and differs from them by at most a factor of 2.

6. In practice, it is well-known that practical networks have very small arboricities (Eppstein et al., 2013). Indeed, in all 39 datasets considered by (Eppstein et al., 2013), the arboricity is at most 201 even though their largest graph has 3.7 million nodes and 16.5 million edges. On 32/39 of their datasets, the arboricity is less than 60. We note that (Eppstein et al., 2013) report the degeneracy, which is an upper bound on the arboricity.

**Relationship to our architecture.** Next, we briefly describe how the Chiba–Nishizeki algorithm inspired our architecture. Recall that the algorithm iterates over all $(u,v)$; then it iterates over all $w \in N(u)$ and checks whether $w \in N(v)$ to see if it should report a triangle. In other words, it only reports triangles for vertices $w$ such that $w \in N(u) \cap N(v)$.

When looking at this procedure from the perspective of distributed computing, one can view this as follows: Each edge $(u,v)$ aggregates the neighborhoods $N(u)$ and $N(v)$ and then computes their intersection $N(u) \cap N(v)$ to obtain in which triangles it appears in.

Indeed, this distributed point of view is the motivation for our Equations (6), (7) and (8): Equation (6) aggregates the embeddings of neighbors $N(u)$ of $u$, Equation (8) aggregates the embeddings of neighbors $N(v)$ of $v$, and Equation (7) aggregates the embeddings of all edges $(u,w)$ and $(v,w)$ such that $w \in N(u) \cap N(v)$ forms a triangle with $u$ and $v$.

## C. Omitted proofs

In this section, we provide missing proofs from the main text.

### C.1. Proof of Proposition 1

This follows immediately from the discussion in the two paragraphs preceding the statement of Proposition 1.

### C.2. Proof of Theorem 2

We show Theorem 2, i.e., that the EB-1-WL test is strictly more expressive than 1-WL and the edge-based NC-1-WL test by Liu et al. (2024) in distinguishing non-isomorphic graphs.

Our *proof idea* is as follows. Looking at the graphs in Figure 4, all nodes of $G$ and $H$ have the same degree and lie in the same number of triangles. Hence, they are not distinguished by NC-1-WL. However, they are distinguished by EB-1-WL since every edge in $H$ is part of two triangles, whereas $G$ has edges that belong to only a single triangle. Interestingly, we also show that EB-1-WL is strictly more expressive than 1-WL with triangle counts added on node and edge level. Furthermore, we note that any pair of graphs distinguishable by EB-1-WL is also distinguishable by 2-WL, though the

reverse implication does not hold. A concrete example of a pair of graphs that is distinguishable by 2-WL but neither by EB-1-WL nor NC-1-WL is shown in Figure 3.

Next, we give the formal proof. For this, we split the theorem into two lemmas and prove them separately.

**Lemma 6.** *Every pair of graphs distinguishably by NC-1-WL is also distinguished by EB-1-WL*

Fix a graph $G = (V, E)$ (we omit $G$ in the notation for colors from now on). To simplify the presentation, we will also use the following notation. An "ordered edge" is an ordered pair of nodes, connected by an edge (EB-1-WL assigns colors to ordered edges). Now, an "ordered triangle" is an ordered triple of nodes where all nodes are connected by an edge.

For example, if $u$ is a node of $G$, then $(u, *)$ is the set of ordered pairs of nodes connected by an edge, where the first node in the pair is $u$. Likewise, $(*, u)$ will be a similar set but for pairs where the second node is $u$.

More generally, if we have a $k$-tuple where some coordinates are nodes of $G$, and some coordinates are $*$ (meaning "undefined"), this tuple denotes the set of all ways to replace $*$'s by nodes of $G$ such that all pairs of nodes in the tuple are connected by an edge. For example, $(*, *, *)$ means the set of all ordered triangles, and $(u, *, *)$ denotes the set of all ordered triangles that have $u$ as the first coordinate.

Next, if $c$ is a coloring of nodes, and $t$ is a tuple of nodes, we will write $c(t)$ for the tuple of colors of nodes in $t$, for instance, if $t = (u, v, w)$, then $c(t) = (c(u), c(v), c(w))$. Likewise, if $c$ is a coloring of pairs of nodes, and $t$ is a tuple, then we will write $c(t)$ for the tuple of colors of all pairs of nodes in $t$. We will use this when $t$ is at most a triple of nodes, where then it is defined as:

$$c(u, v, w) = (c(u, v), c(u, w), c(v, w)).$$

Moreover, we extend this notation to tuples with $*$s. Namely, if $T$ is a tuple with stars (a set of tuples of nodes without stars), then $c(T) = \{\!\{ c(t) \mid t \in T \}\!\}$.

In this notation, the updates of NC-1-WL and EB-1-WL can be defined as follows. The color $nc^{(\ell+1)}(u)$ is defined by multisets $nc^{(\ell+1)}(u, *)$ and $nc^{(\ell+1)}(u, *, *)$. In turn, the color $\mathsf{eb}^{(\ell+1)}(u, v)$ is defined by $\mathsf{eb}^{(\ell)}(u, *)$, $\mathsf{eb}^{(\ell)}(v, *)$, and $\mathsf{eb}^{(\ell)}(u, v, *)$.

To establish the claim, it is enough to show that $\mathsf{eb}(*, *)$ uniquely determines $\mathsf{nc}^{(\ell)}(*)$ for every $\ell \geq 0$.

**Lemma 7** (Technical version of Lemma 6). *For every $\ell \geq 0$, and for every ordered edge $(u, v)$, we have that $\mathsf{eb}(u, v)$ uniquely determines $\mathsf{nc}^{(\ell)}(u)$ and $\mathsf{nc}^{(\ell)}(v)$.*

*Proof.* The proof is by induction on $\ell$. For $\ell = 0$, all nodes have the same $\mathsf{nc}^{(0)}$-color, so there is nothing to prove. Assume now that the statement is proved for $\ell$, we establish it for $\ell + 1$. That is, we have to show that $\mathsf{eb}(u, v)$ uniquely determines $\mathsf{nc}^{(\ell+1)}(u)$ and $\mathsf{nc}^{(\ell+1)}(v)$. Without loss of generality, we only show that $\mathsf{eb}(u, v)$ determines $\mathsf{nc}^{(\ell+1)}(u)$.

By definition, $\mathsf{eb}^{(\ell)}(u, v)$ uniquely determines $\mathsf{eb}^{(\ell-1)}(u, *)$ and $\mathsf{eb}^{(\ell-1)}(u, v, *)$. If we make one more step, from $\mathsf{eb}^{(\ell-1)}(u, *)$ we can determine $\mathsf{eb}^{(\ell-2)}(u, *, *)$. Indeed, we use the fact that we can determine $\mathsf{eb}^{(\ell-2)}(u, w, *)$ from $\mathsf{eb}^{(\ell-1)}(u, w)$ for $(u, w) \in (u, *)$.

For a stable EB-1-WL coloring eb, we thus get that $\mathsf{eb}(u, v)$ uniquely determines $\mathsf{eb}(u, *)$ and $\mathsf{eb}(u, *, *)$. They, in turn, by the induction hypothesis, uniquely determined $\mathsf{nc}^{\ell}(u, *)$ and $\mathsf{nc}^{\ell}(u, *, *)$. Thus, from this information, we get $\mathsf{nc}^{(\ell+1)}(u)$, as required.

We now finish the proof of the claim. Assume that we want to know how many times a node color $c$ appears in $\mathsf{nc}^{(\ell)}(*)$. We can assume that $\ell \geq 1$ (we can determine the multiset for $\ell = 0$ from the multiset for $\ell = 1$), then the color $c$ uniquely determines the degree $d$ of a node (which is not 0 by the assumption of the absence of isolated nodes). We go through all ordered edges $(u, v)$ and count how many times we have $\mathsf{nc}^{(\ell)}(u) = c$. We count every node $d$ times, so we have to divide this number by $d$.

$\square$

We move on to the second part of the theorem.

**Lemma 8.** *There exist graphs that are distinguished by EB-1-WL but not by NC-1-WL.*

Graph $G$        Graph $H$

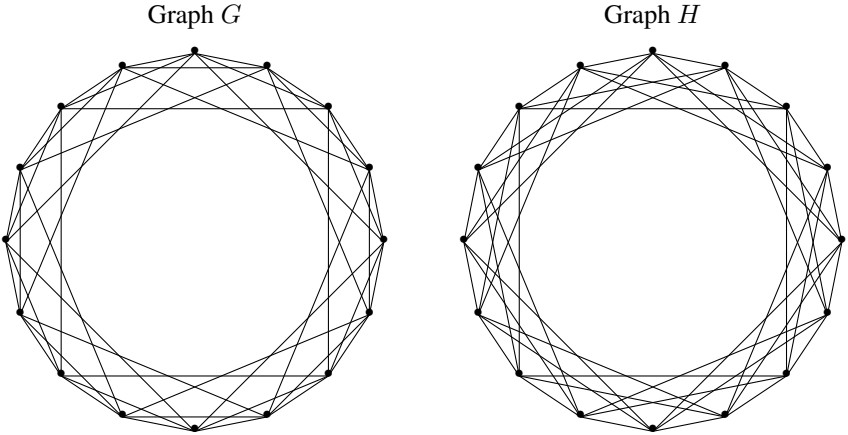

A pair of graphs that EB-1-WL can distinguish, but NC-1-WL cannot.

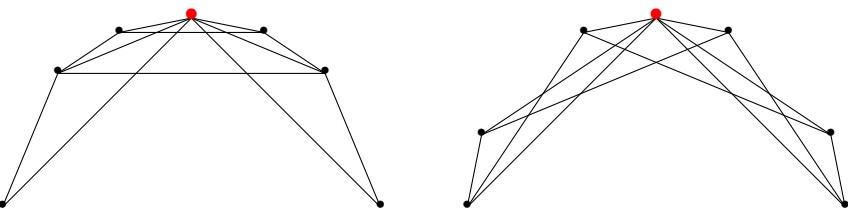

In both graphs, every node has the same degree and is part of 6 triangles; here shown for the top node ( ■ ). Hence, NC-1-WL cannot distinguish this pair of graphs.

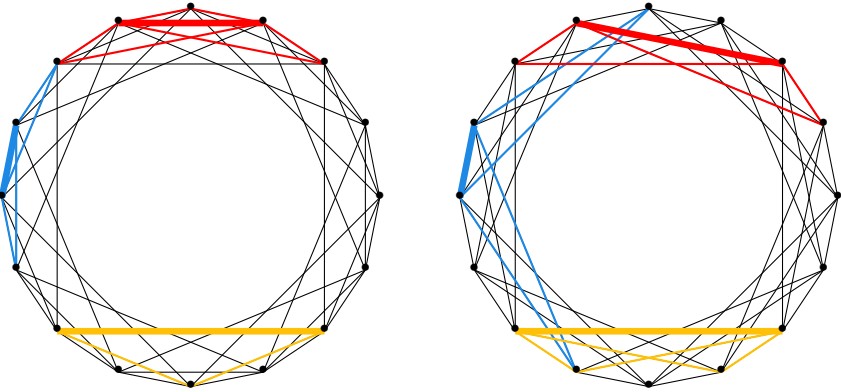

Each thick colored edge represents one of the three edge types, together with all triangles it is a part of. We can see that all edges in $H$ are in exactly two triangles whereas this is not the case for $G$ (see ■). Hence, EB-GNN can distinguish this graph pair.

*Figure 4.* Visual proof of Lemma 8 which is used to prove Theorem 2.

*Proof.* This follows directly from Figure 4. This figure shows a graph that is indistinguishable by NC-1-WL which follows from the fact that every node in both graphs has the same degree and is part of 6 triangles. However, in graph $G$ there exist edges which are only in one triangle whereas in $H$ every edge is in exactly two triangles. Hence, EB-GNN can distinguish this graph pair. □

### C.3. Proof of Theorem 3

We start by showing the following.

**Lemma 9.** *For each possible color $c$ that the EB-1WL test obtains after $t$ rounds, there is a formula $\phi_c(x, y)$ of quantifier depth $t$ in CFOC$^3$ such that $\mathsf{eb}^t(G, (u, v)) = c$ if and only if $G \models \phi_c(u, v)$, for each graph $G = (V, E)$ and ordered tuple $(u, v)$ such that $\{u, v\} \in E$.*

*Proof.* We prove the result by induction on the number of rounds $t \geq 0$. By definition of EB-1-WL, before any refinement every edge $\{u, v\}$ receives the same initial color $c = 1$. Accordingly, we can define $\phi_c(x, y) := E(x, y)$.

Suppose we already have, for every color $d$ occurring after $t$ rounds of EB-1-WL, a formula $\phi_d(x, y)$ that defines it. Consider now a new color $c$ obtained after $t + 1$ rounds. By the definition of EB-1-WL, the color of an edge $\{u, v\}$ at step $t + 1$ is completely determined by the following information from step $t$:

1. the color $d$ previously assigned to $\{u, v\}$,
2. the multiset $\mathcal{E}$ of colors of edges of the form $\{u, w\}$,
3. the multiset $\mathcal{G}$ of colors of edges of the form $\{v, w\}$, and
4. the multiset $\mathcal{F}$ of pairs of colors $(f_1, f_2)$ corresponding to edges $\{u, w\}$ and $\{v, w\}$ incident with a common neighbor $w$.

It follows that the new color $c$ can be described in CFOC$^3$ by the formula:

$$\phi_c(x, y) := \phi_d(x, y) \land$$

$$\bigwedge_{e \in \mathcal{E}} \exists^{=n_e} z \big( E(x, z) \land \phi_e(x, z) \big) \land \exists^{=\sum_{e \in \mathcal{E}} n_e} z E(x, z)$$

$$\bigwedge_{g \in \mathcal{G}} \exists^{=m_g} z \big( E(z, y) \land \phi_g(z, y) \big) \land \exists^{=\sum_{g \in \mathcal{G}} m_g} z E(y, z)$$

$$\bigwedge_{(f_1, f_2) \in \mathcal{F}} \exists^{=p_{(f_1, f_2)}} z \big( E(x, z) \land E(y, z) \land \phi_{f_1}(x, z) \land \phi_{f_2}(y, z) \big) \land$$

$$\exists^{=\sum_{(f_1, f_2) \in \mathcal{F}} p_{(f_1, f_2)}} z \big( E(x, z) \land E(y, z) \big).$$

Here, $n_e$ is the multiplicity of $e$ in $\mathcal{E}$, $m_g$ the multiplicity of $g$ in $\mathcal{G}$, and $p_{(f_1, f_2)}$ the multiplicity of $(f_1, f_2)$ in $\mathcal{F}$. As usual, we write $\exists^{=n} x \, \psi$ as shorthand for $\exists^{\geq n} x \, \psi \land \neg \exists^{\geq n+1} x \, \psi$. Notice that $\phi_c(x, y)$ has quantifier depth $t + 1$. □

Let $G = (V, E)$ be a graph and $\bar{v}$ a tuple of elements in $V$. For $t \geq 0$, we write $\mathsf{tp}^t(G, \bar{v})$ for the *$t$-type of* $(G, \bar{v})$, which is the set of formulas $\phi(\bar{x})$ with quantifier depth $t$ in CFOC$^3$ such that $G \models \phi(\bar{v})$. When $\bar{v}$ is the empty tuple, we simply write $\mathsf{tp}^t(G)$ to denote the set of all CFOC$^3$ sentences (formulas with no free variables) of quantifier depth $t$ that hold true in $G$. Also, $\mathsf{eb}^t(G)$ is the multiset formed by all elements of the form $\mathsf{eb}^t(G, (u, v))$, where $\{u, v\} \in E$.

**Lemma 10.** *Consider graphs $G = (V, E)$ and $G' = (V', E')$ and edges $\{u, v\} \in E$ and $\{u', v'\} \in E'$. Then for each $t \geq 0$, the following hold:*

*(1) If $\mathsf{eb}^t(G) = \mathsf{eb}^t(G')$ then $\mathsf{tp}^t(G) = \mathsf{tp}^t(G')$.*

*(2) If $\mathsf{eb}^t(G, (u, v)) = \mathsf{eb}^t(G', (u', v'))$ and $\mathsf{eb}^t(G) = \mathsf{eb}^t(G')$, then $\mathsf{tp}^t(G, u) = \mathsf{tp}^t(G', u')$, $\mathsf{tp}^t(G, v) = \mathsf{tp}^t(G', v')$, and $\mathsf{tp}^t(G, (u, v)) = \mathsf{tp}^t(G', (u', v'))$.*

*Proof.* We prove this by induction on $t \geq 0$. The base case $t = 0$ holds trivially. In fact, there are no sentences in CFOC$^3$ of quantifier depth 0, and hence $\mathsf{tp}^0(G) = \mathsf{tp}^0(G')$ holds vacuously. Moreover, any formula $\phi(x)$ of quantifier depth 0 with only one free variable $x$ in CFOC$^3$ is a Boolean combination of formulas of the form $E(x, x)$. Since both $G \models \neg E(u, u)$ and $G' \models \neg E(u', u')$, it is the case that $G \models \phi(u) \Leftrightarrow G' \models \phi(u')$. The same holds for $v$. Finally, any formula $\phi(x, y)$ of

quantifier depth $0$ with only two free variables $x$ and $y$ in $\text{CFOC}^3$ is a Boolean combination of formulas of the form $E(x, y)$, $E(x, x)$, and $E(y, y)$. Since both $G \models \neg E(u, u)$ and $G' \models \neg E(u', u')$, both $G \models \neg E(v, v)$ and $G' \models \neg E(v', v')$, and both $G \models E(u, v)$ and $G' \models E(u', v')$, it is the case that $G \models \phi(u, v) \Leftrightarrow G' \models \phi(u', v')$.

Let us consider now the inductive case $t + 1$, for $t \geq 0$. We prove (1) and (2) separately.

- We first prove **(1)**. For this, we assume that $\text{eb}^{t+1}(G) = \text{eb}^{t+1}(G')$ and show

$$G \models \phi \iff G' \models \phi$$

for an arbitrary Boolean $\text{CFOC}^3$ sentence $\phi$ of quantifier depth $t+1$. From the quantifier depth of $\phi$ it follows that it has form $\exists^{\geq n} x\, \alpha(x)$, where $\alpha(x)$ has quantifier depth $t$. Since $\alpha$ has quantifier depth $t$ we can prove $G \models \phi \iff G' \models \phi$ by showing

$$\{\{\text{tp}^t(G, u) \mid u \in V\}\} = \{\{\text{tp}^t(G', u') \mid u' \in V'\}\}. \tag{12}$$

Without loss of generality, we take an arbitrary $\tau \in \text{tp}^t(G, u)$ for some $u \in V$. We prove Equation (12) by showing that in both graphs the same number of vertices satisfy formula $\tau$. Since $t > 0$, every vertex with $\tau \in \text{tp}^t(G, v)$ must have the same degree $d$, because they satisfy the same formulas of the form $\exists^{\geq n} y\, E(x, y)$.
Our assumption $\text{eb}^{t+1}(G) = \text{eb}^{t+1}(G')$ implies

$$\{\{\text{eb}^t(G, (u, v)) \mid \{u, v\} \in E\}\} = \{\{\text{eb}^t(G', (u', v')) \mid \{u', v'\} \in E'\}\},$$

and therefore, by the induction hypothesis,

$$\{\{\text{tp}^t(G, (u, v)) \mid \{u, v\} \in E\}\} = \{\{\text{tp}^t(G', (u', v')) \mid \{u', v'\} \in E'\}\}. \tag{13}$$

Moreover, $\text{tp}^t(G, (u, v))$ determines both $\text{tp}^t(G, u)$ and $\text{tp}^t(G, v)$. Define

$$S(G, \tau) = \{\{\text{tp}^t(G, (u, v)) \mid \{u, v\} \in E \text{ and } \text{tp}^t(G, u) = \tau\}\}.$$

Then, the number of vertices $v \in V$ and $v' \in V'$ that satisfy $\tau$ are

$$|S(G, \tau)|/d \quad \text{and} \quad |S(G', \tau)|/d,$$

respectively. Because $|S(G, \tau)| = |S(G', \tau)|$ by Equation (13), the two counts coincide, which completes the proof.
- We now prove **(2)**. Assume that $\text{eb}^{t+1}(G, (u, v)) = \text{eb}^{t+1}(G', (u', v'))$ and $\text{eb}^{t+1}(G) = \text{eb}^{t+1}(G')$. We only show that $\text{tp}^{t+1}(G, (u, v)) = \text{tp}^{t+1}(G', (u', v'))$, as the remaining cases are conceptually analogous. Consider a formula $\phi(x, y)$ in $\text{CFOC}^3$ of quantifier depth $t + 1$. Then $\phi(x, y)$ must be of the form $E(x, y) \wedge \alpha(x, y)$, where $\alpha(x, y)$ is a $\text{CFOC}^3$ formula of quantifier depth $t + 1$. Since both $G \models E(u, v)$ and $G' \models E(u', v')$, we only have to show that $G \models \alpha(u, v) \Leftrightarrow G \models \alpha(u', v')$. By definition, $\alpha(x, y)$ is a Boolean combination of:
  (a) formulas $\beta(x, y)$ of quantifier depth $t + 1$ with two free variables, $x$ and $y$,
  (b) formulas $\gamma(x)$ of quantifier depth $t + 1$ with only one free variable, $x$,
  (c) formulas $\delta(y)$ of quantifier depth $t + 1$ with only one free variable, $y$, and
  (d) sentences $\eta$ of quantifier depth $t + 1$.

**Case (a):** Consider first a formula as above of the form $\beta(x, y)$. By construction, $\beta(x, y) = E(x, y) \wedge \exists^{\geq n} z\, (E(x, z) \wedge E(y, z) \wedge \beta_1(x, y, z))$, where $\beta_1(x, y, z)$ is a $\text{CFOC}^3$ formula of quantifier depth $t$ which must be a Boolean combination of other formulas of quantifier depth $t$. Recall, that our logic $\text{CFOC}^3$ only allows formulas to contain three variables. However, we need one variable to increase the quantifier depth. Hence, all subformulas of $\beta_1$ can have at most two free variables from the set $\{x, y, z\}$.
By assumption, $\text{eb}^{t+1}(G, (u, v)) = \text{eb}^{t+1}(G', (u', v'))$, and hence both $\text{eb}^t(G, (u, v)) = \text{eb}^t(G', (u', v'))$ and

$$\{\{(\text{eb}^t(G, (u, w)), \text{eb}^t(G, (v, w))) \mid \{u, w\}, \{v, w\} \in E\}\} =$$
$$\{\{(\text{eb}^t(G', (u', w')), \text{eb}^t(G', (v', w'))) \mid \{u', w'\}, \{v', w'\} \in E'\}\}.$$

Also by assumption, $\text{eb}^{t+1}(G) = \text{eb}^{t+1}(G')$, which implies that $\text{eb}^t(G) = \text{eb}^t(G')$.

By the induction hypothesis, we conclude that $\mathsf{tp}^t(G, (u, v)) = \mathsf{tp}^t(G', (u', v'))$ and:

$$\{\{(\mathsf{tp}^t(G, (u, w)), \mathsf{tp}^t(G, (v, w))) \mid \{u, w\}, \{v, w\} \in E\}\} =$$
$$\{\{((\mathsf{tp}^t(G', (u', w')), \mathsf{tp}^t(G', (v', w'))) \mid \{u', w'\}, \{v', w'\} \in E'\}\}.$$

Since $\beta_1$ has quantifier depth $t$, this implies that

$$|\{w \in N(u) \cap N(v) \mid G \models \beta_1(u, v, w)\}| = |\{w' \in N(u') \cap N(v') \mid G' \models \beta_1(u', v', w')\}|,$$

which means that $G \models \beta(u, v) \Leftrightarrow G' \models \beta(u', v')$.

**Case (b):** Consider second a formula as above of the form $\gamma(x)$. By construction, $\gamma(x) = \exists^{\geq n} z\, (E(x, z) \wedge \gamma_1(x, z))$, where $\gamma_1(x, z)$ is a CFOC$^3$ formula of quantifier depth $t$. By assumption, $\mathsf{eb}^{t+1}(G, (u, v)) = \mathsf{eb}^{t+1}(G', (u', v'))$, and hence

$$\{\{\mathsf{eb}^t(G, (u, w)) \mid \{u, w\} \in E\}\} = \{\{\mathsf{eb}^t(G', (u', w')) \mid \{u', w'\} \in E'\}\}.$$

By induction hypothesis, this implies that:

$$\{\{\mathsf{tp}^t(G, (u, w)) \mid \{u, w\} \in E\}\} = \{\{\mathsf{tp}^t(G', (u', w')) \mid \{u', w'\} \in E'\}\}.$$

By focusing on those types that contain $\gamma_1(x, z)$, we obtain:

$$|\{w \in N(u) \mid G \models \gamma_1(u, w)\}| = |\{w' \in N(u') \mid G' \models \gamma_1(u', w')\}|,$$

which means that $G \models \gamma(u) \Leftrightarrow G' \models \gamma(u')$.

**Cases (c) and (d):** The formulas of the form $\delta(y)$ are handled analogously to the previous case. Finally, consider a sentence as above of the form $\eta$. Since $\mathsf{eb}^{t+1}(G) = \mathsf{eb}^{t+1}(G')$, we have by part (1) that $G \models \eta \Leftrightarrow G' \models \eta$.

This finishes the proof of the lemma. $\qquad\square$

We now prove Theorem 3.

*Proof of Theorem 3.* Assume first that $\mathsf{eb}(G) = \mathsf{eb}(G')$, for graphs $G$ and $G'$. In particular, then, $\mathsf{eb}^t(G) = \mathsf{eb}^t(G')$ for every $t \geq 0$. Take an arbitrary sentence $\phi$ of CFOC$^3$, and assume that its quantifier depth is $t$. From Lemma 10, we conclude that $\mathsf{tp}^t(G) = \mathsf{tp}^t(G')$, and hence $G \models \phi \Leftrightarrow G' \models \phi'$.

Assume, on the contrary, that $G \models \phi \Leftrightarrow G \models \phi'$, for every CFOC$^3$ sentence $\phi$. Hence, $\mathsf{tp}^t(G) = \mathsf{tp}^t(G')$ for every $t \geq 0$. To prove $\mathsf{eb}(G) = \mathsf{eb}(G')$ it suffices to show $\mathsf{eb}^t(G) = \mathsf{eb}^t(G')$ for each $t \geq 0$. For a node $v \in V$, define $\mathsf{eb}^t(v) = \{\{\mathsf{eb}^t(v, w) \mid \{v, w\} \in E\}\}$. Notice, then, that $\mathsf{eb}^t(G)$ is completely determined by the multiset $\Gamma(G) = \{\{\mathsf{eb}^t(v) \mid v \in V\}\}$. For each element $\tau \in \Gamma(G)$, we write $\mathcal{C}_\tau$ for the multiset of colors computed by EB-1-WL after $t$ steps that belong to $\tau$.

Define the following formula from CFOC$^3$:

$$\phi_G := \Big( \bigwedge_{\tau \in \Gamma(G)} \exists^{=\ell_\tau} x\, \phi_\tau(x) \Big) \wedge \exists^{=\sum_{\tau \in \Gamma(G)} \ell_\tau} x(x = x),$$

where $\ell_\tau$ is the multiplicity of $\tau$ in $\Gamma(G)$ and $\phi_\tau(x)$ is defined as follows:

$$\phi_\tau(x) = \Big( \bigwedge_{c \in \mathcal{C}_\tau} \exists^{=q_c} y\, \big(E(x, y) \wedge \phi_c(x, y)\big) \Big) \wedge \exists^{=\sum_{c \in \mathcal{C}_\tau} q_c} y\, E(x, y),$$

where $q_\tau$ is the cardinality of $\tau$ in $\Gamma(G)$ and $\phi_c(x, y)$ is the CFOC$^3$ formula from Lemma 9. Notice that $\phi_G$ has quantifier depth bounded by $t + 2$.

It is easy to see that $G' \models \phi_G \Leftrightarrow \Gamma(G) = \Gamma(G')$. Since $G \models \phi_G$ and $\mathsf{tp}^{t+2}(G) = \mathsf{tp}^{t+2}(G')$, we conclude that $G' \models \phi_G$, and hence $\Gamma(G) = \Gamma(G')$. Since $\Gamma(G)$ determines $\mathsf{eb}^t(G)$, we conclude that $\mathsf{eb}^t(G) = \mathsf{eb}^t(G')$. $\qquad\square$

### C.4. Proof of Theorem 4

It is well-known that chordal graphs admit a *perfect elimination order*: an ordering of its nodes such that every node $v$ and its neighbors that go before $v$ in the order form a clique (Rose, 1970). Graphs of tree-width 2 cannot have cliques larger than a triangle; this means that for any chordal graph $H$ of tree-width 2 there exists an ordering $v_1, \ldots, v_m$ of its nodes such that for any $k \in \{1, \ldots, m\}$, one of the following holds:

- (a) $v_k$ is not connected to any node out of $v_1, \ldots, v_{k-1}$;
- (b) $v_k$ is connected to exactly one node out of $v_1, \ldots, v_{k-1}$;
- (c) $v_k$ is connected to exactly two nodes $v_i, v_j \in \{v_1, \ldots, v_{k-1}\}$, and $v_i$ and $v_j$ are also connected.

Let eb be the stable EB-1WL coloring of $G$. Let us show that the multiset $\mathsf{eb}(G)$ of eb-labels of ordered edges uniquely determines the number of homomorphisms from $H$ to $G$, for any tree-width 2 chordal graph $H$. This means that if two graphs $G_1$ and $G_2$ have a different number of homomorphisms from some graph $H$ like that, they will be distinguished by the EB-1WL on the stage where colorings of both graphs stabilize.

For a node $u$, define:

$$\mathsf{eb}(u) = \{\!\!\{\,\mathsf{eb}(u, w) \mid w \in N(u)\,\}\!\!\}.$$

Note that $\mathsf{eb}(u, v)$, as it is stable, uniquely determines induced labels of $u$ and $v$ through (2) and (4). Moreover, $\mathsf{eb}(G)$ uniquely determines the multiset $\mathsf{eb}(V) = \{\!\!\{\,\mathsf{eb}(u) \mid u \in V\,\}\!\!\}$. Namely, we go through all $\mathsf{eb}(u, v)$, compute $\mathsf{eb}(u)$ from it, which in turn determines the degree of $u$. We then divide the number of occurrences of $\mathsf{eb}(u)$ by the degree.

Consider any labeling of edges of $H$ by labels from $\mathsf{eb}(G)$, and of its nodes by labels from $\mathsf{eb}(V)$. We show that for any such labeling, the number of homomorphisms from $H$ to $G$ that "preserve" this labeling is determined just by the multiset $\mathsf{eb}(G)$. The total number of homomorphisms is hence also determined by $\mathsf{eb}(G)$, since we can go through all possible labelings of $H$ and sum up homomorphisms for all of them.

Here, "preserve" formally means that (a) a node $v_j$ with a label $\ell$ goes into a node in $G$ that has this eb-label (b) if $v_i, v_j$ is an edge of $H$ (where $i < j$ are indices of these nodes in the perfect elimination order), and if $v_i$ and $v_j$ go to some nodes $u_i, u_j$ in $G$, respectively, then the label of the edge $v_i, v_j$ in $H$ has to be equal to $\mathsf{eb}(u_i, u_j)$.

We show that we can compute the number of homomorphisms that preserve a given labeling of $H$, just knowing $\mathsf{eb}(G)$, by first computing how many ways we can define the image of $v_1$, then the image of $v_2$, then of $v_3$, and so on.

As for $v_1$, it is assigned a label $\ell$ in the labeling; we know the multiset of eb-labels of the nodes of $G$, which determines the number of ways we can define the image of $v_1$ (this is the number of nodes of $G$ that have label $\ell$).

Now, assume that we have defined images of $v_1, \ldots, v_{k-1}$. Now, it's $v_k$'s turn. Firstly, it is possible that $v_k$ is not connected to any node among $v_1, \ldots, v_{k-1}$. Then we can freely map $v_k$ to any node of $G$ that has the same label $\ell$ as assigned to $v_k$ in the coloring. We just have to multiply the current number of homomorphisms by the number of nodes in $G$ with this label $\ell$.

If $v_k$ is connected to a single node $v_i$, $i < k$, and $v_i$ is already mapped to some node $u_i$, then we have to map $v_k$ to some node $u_k$ that is connected to $u_i$ and such that the $\mathsf{eb}(u_i, u_k)$ coincides with the color of the edge $v_i, v_k$ in $H$ (this color also determines the label of $u_k$ which has to be consistent with the label that $v_k$ has in the labeling of $H$, otherwise the number of homomorphisms is just 0). The number of ways to choose such $u_k$ is thus determined by $\{\!\!\{\,\mathsf{eb}(u_i, w) \mid w \in N(u_i)\,\}\!\!\} = \mathsf{eb}(u_i)$, which in turn equals the label of $v_i$ in the labeling of $H$. We multiply the current number of homomorphisms by the number of occurrences of the label of the edge $v_i, v_k$ in the label of $v_i$.

The same argument holds for the third case when $v_k$ is connected to previous nodes $v_i, v_j$ that are connected by an edge. The number of ways to choose the image of $v_k$ is the number of occurrences of the pair $(\ell_{ik}, \ell_{jk})$ in (3) in the label of the edge $\ell_{ij}$, where $\ell_{ik}, \ell_{jk}$, and $\ell_{ij}$ are labels of edges $v_i, v_k$, $v_j, v_k$, and $v_i, v_j$ in $H$, respectively.

### C.5. Proof of Theorem 5

In our proof, we show that EB-GNNs require dimension $O(m + t)$, where $m$ is the number of edges and $t$ the number of triangles. Whether this can be reduced to $O(\log n)$, as shown for simulating 1WL by MPNNs (Aamand et al., 2022), remains open. Moreover, by replacing ReLU with any analytic non-polynomial activation and concatenation in Eq. (7) with a Hadamard product, one can prove Thm. 5 already holds for $d = 1$, following techniques from Amir et al. (2023), Bravo et al. (2024), and Hordan et al. (2024).

The theorem is deduced from the following lemma.

**Lemma 11.** *Let $f_1, \ldots, f_n \in \mathbb{R}^d$ be $n$ distinct vectors. Then there exists a matrix $A \in \mathbb{R}^{n \times d}$ and a vector $b \in \mathbb{R}^n$ such that the vectors*

$$g_1 = \mathrm{ReLU}(Af_1 + b), \ldots, g_n = \mathrm{ReLU}(Af_n + b)$$

*are linearly independent.*

Indeed, consider any two graphs $G$ and $H$, distinguishable by the EB-1WL test. We construct an EB GNN $\mathcal{T}$ that distinguishes $G$ and $H$.

Consider the disjoint union of $G$ and $H$. Let $m$ be the number of ordered edges of this union, and $t$ be the number of ordered triangles. The dimension of $\mathcal{T}$ will be

$$d = 1 + 2m + t.$$

We show that there exists a choice of parameter matrices such that, for any $i$, a) EB-1WL labels after $i$ iterations are in a one-to-one correspondence with feature vectors $f^{(i)}(u, v)$; b) all coordinates of $f^{(i)}$, except the first one, are 0s.

For $i = 0$, this holds because all edges initially have the same EB-1WL label, and because of (5).

Assume that it holds after $i - 1$ iterations. We use Lemma 11 in (6–8) to map distinct feature vectors $f^{(i-1)}(u, v)$ (there are at most $m$ of them) or their pairs as in (7)) (there are at most $t$ such pairs) to linearly independent vectors. This ensures that we obtain different sums for different multisets of terms in (6–8). We can use 3 blocks of $m, m$ and $t$ disjoint coordinates that are "'free'" in vectors $f^{(i-1)}(u, v)$ so that the sum in (9) uniquely determines the whole 4-tuple in the definition of the updated EB-1WL feature.

We can then define a feed-forward network to injectively map vectors $g^{(i)}(u, v)$ to vectors $f^{(i)}(u, v)$ that have all coordinates, except the first one, equal to 0. There exists a vector $w = (w_1, \ldots, w_d) \in \mathbb{R}^d$ such that $\langle w, g^{(i)}(u, v) \rangle \neq \langle w, g^{(i)}(u', v') \rangle$ whenever $g^{(i)}(u, v) \neq g^{(i)}(u', v')$ (for each of the finitely many pairs of distinct $g^{(i)}$-vectors, the set of $w$ for which we have equality is a hyperplane in $\mathbb{R}^d$). Hence, it is enough to realize the following linear transformation by a FFN:

$$g^{(i)}(u, v) \mapsto \begin{pmatrix} w_1 & w_2 & \ldots & w_d \\ 0 & 0 & \ldots & 0 \\ \vdots & \vdots & \ddots & \vdots \\ 0 & 0 & \ldots & 0 \end{pmatrix} g^{(i)}(u, v) = W g^{(i)}(u, v).$$

This can be achieved by the following FFN:

$$x \mapsto W \mathrm{ReLU}(x + u_i) - W u_i,$$

where $u_i \in \mathbb{R}^d$ is an arbitrary vector such that $u_i \geq g^{(i)}(u, v)$ coordinate-wise for every ordered edge $(u, v)$. Indeed, then we obtain:

$$x \mapsto W \mathrm{ReLU}(x + u_i) - W u_i = W(x + u_i) - W u_i = W x$$

for any $x$ of the form $x = g^{(i)}(u, v)$, as required.

Now, assume that we are at an iteration when $G$ and $H$ have different multisets of EB-1WL labels. We now need to do one more iteration that maps distinct feature vectors into linearly independent vectors so that $G$ and $H$ will have different final representations in (11). We set all matrices and bias vectors in (6–8) to 0 so that $g^{(i)}(u, v) = f^{(i-1)}(u, v)$. We then use Lemma 11 again to construct a matrix $U$ and a vector $u$ such that the following transformation:

$$f^{(i-1)}(u, v) \mapsto \mathrm{ReLU}(U f^{(i-1)}(u, v) + v) \tag{14}$$

maps distinct vectors into linearly independent ones. There are at most $m$ distinct feature vectors, which means we can use coordinates that are 0 in $g^{(i)}(u, v)$ (by the invariant b)) for the image of (14) so it does not interfere with the first coordinate of $g^{(i)}(u, v)$ in (10) . By setting $V = Id, v = 0$, we obtain a FFN that realizes this transformation.

*Proof of Lemma 11.* Since $f_1, \ldots, f_n$ are distinct, there exists $w \in \mathbb{R}^d$ such that $\langle f_1, w \rangle, \ldots, \langle f_n, w \rangle$ are distinct (because the set of $w$ such that $\langle f_i, w \rangle = \langle f_j, w \rangle$ for some $i \neq j$ is a union of finitely many hyperplanes, not covering the whole $\mathbb{R}^d$). Without loss of generality,

$$\langle f_1, w \rangle < \langle f_2, w \rangle < \ldots < \langle f_n, w \rangle.$$

For $i = 1, \ldots, n$, let $\gamma_i$ be some number between $\langle f_{i-1}, w \rangle$ and $\langle f_i, w \rangle$ (for $i = 1$, this is some number smaller than $\langle f_1, w \rangle$). Define

$$A = \begin{pmatrix} w \\ \vdots \\ w \end{pmatrix}, \qquad b = \begin{pmatrix} -\gamma_1 \\ -\gamma_2 \\ \vdots \\ -\gamma_n \end{pmatrix}.$$

Observe that in the vector

$$Af_i + b = \begin{pmatrix} \langle f_i, w \rangle - \gamma_1 \\ \langle f_i, w \rangle - \gamma_2 \\ \vdots \\ \langle f_i, w \rangle - \gamma_n \end{pmatrix}$$

the first $i$ coordinates are strictly positive, and the other coordinates are strictly negative. Hence, the vector $g_i = \mathrm{ReLU}(Af_i + b)$ is a vector where the first $i$ coordinates are strictly positive, and the rest are 0s. Therefore, $g_1, \ldots, g_n$ are linearly independent. $\qquad\square$

This finishes the proof of the theorem.

## D. More Details on Experiments

We provide additional information on our experimental procedure and more detailed results.

**Model.**    We have defined graph as purely existing of nodes and edges $G = (V, E)$ . However, real-world datasets often use node features and edge features to encode additional information in the graph for all. For every directed edge $(u, v) \in E$, we incorporate the node features of $X_u, X_v$ of $u, v$ and the edge features $W_{(u,v)}$ of $(u, v)$ into our model by initializing the edge embedding $\mathcal{T}^{(0)}(G, (u, v))$ with them. We use three different embedding encoders ENC to map both the node features and the edge features to the embedding dimension of our GNN

$$\mathcal{T}^{(0)}(G, (u, v)) = \mathrm{ENC}_{\mathrm{left}}(X_u) + \mathrm{ENC}_{\mathrm{right}}(X_v) + \mathrm{ENC}_{\mathrm{edge}}\left(W_{(u,v)}\right).$$

After obtaining the initial edge embedding, we perform multiple iterations of edge based message passing. After final iteration $t$, we pool edge embeddings into the shape required by the task. For graph-level tasks, we experiment with three methods. We compute graph-level embeddings by either summing

$$\mathcal{T}_{\mathrm{SUM}}(G) = \sum_{(u,v):\{u,v\} \in E} \mathcal{T}^t(G, (u, v)),$$

computing the mean, or computing the mean scaled by the number of nodes (mimicking sum pooling in MPNNs)

$$\mathcal{T}_{\mathrm{MEAN}}(G) = \frac{1}{|E|}\mathcal{T}_{\mathrm{SUM}}(G), \qquad \mathcal{T}_{\mathrm{NODESUM}}(G) = \frac{|V|}{|E|}\mathcal{T}_{\mathrm{SUM}}(G).$$

For edge-level tasks, note that we compute embeddings of directed edges whereas tasks we worked on were on *undirected* regression edges. We experiment with two methods of performing undirected edge predictions. First, we simply sum the embedding of the two directed edges and perform a prediction on this combined embedding. Second, we make a prediction for both directions and combine these predictions by computing the mean.

All our models were implemented in PyTorch Geometric (Paszke et al., 2019; Fey & Lenssen, 2019). All our models were trained on servers with one NVIDIA GeForce RTX 3080 GPU (10 GB VRAM) and 64 GB of RAM. When training a model we evaluate its performance after every epoch on the validation and test set. This performance is measured in the metric that is most commonly used on that dataset. After training, we report the validation and test performance in the epoch with the best validation performance. For real-life datasets, we perform hyperparameter tuning where we pick the hyperparameter combination based on the best validation performance. We train a model with this hyperparameter combination multiple times on different seeds reporting the mean and standard deviation of the test metric.

In general, all our models are trained with a Cosine learning rate scheduler and a learning rate of 0.001 (except on `BREC` where we use the procedure provided by Wang & Zhang (2024)). The final prediction is made by a two-layer MLP. On all real-life datasets EB-GNN uses both skip connections and feed-forward layers, but not on the synthetic datasets `CSL` and `BREC`. Below, we discuss the different setup we used for each dataset. For more details, please consider our code at https://github.com/ocatias/EdgeBasedGNNs.

**CSL.** We train all models for 1000 epochs with a Cosine learning rate scheduler. All GNNs have 5 layers and an embedding dimension of 64.

**BREC.** We train and evaluate our model with the procedure provided by Wang & Zhang (2024). Our EB-GNN model has 10 layers, an embedding dimension of 16 and uses some pooling. We have observed that using the output of Eq. 9 instead of Eq. 10 as edge embeddings leads to better results on the extension graphs. We believe that this is due to numerical issues caused by the large number of layers and small embedding dimension (which is necessary to fit the model into GPU memory). Our MPNN models also have 10 layers but use an embedding dimension of 64.

**QMD.** We train separate models for each of the 8 different tasks. We tune the hyperparameters of EB-GNN for each task based on the grid in Table 5. We train for 500 epochs with a batch size of 1024 and evaluate on 10 different seeds.

**QM9.** We train separate models for each of the 11 different tasks. Initially, we repeated the same procedure as for `QMD`. However, training with a significantly larger batch size than models in literature might harm our performance. Thus, after the initial hyperparameter sweep (Table 5) we used the best hyperparameters and additionally tuned the batch size together with another pooling operation (Tab. 6). We evaluate the best hyperparameter combination on 10 different seeds.

**MalNet-Tiny.** As the graphs in `MalNet-Tiny` are very large, we did not perform any hyperparameter tuning. Our model uses mean pooling, has 5 message passing layers and an embedding dimension of 64. It is trained for 500 epochs with a batch size of 16 and evaluated on 5 different seeds. Note that this graph is *directed*. For this we extended EB-GNN as follows. We only compute edge embeddings for directed edges that are part of the graph. Furthermore, for $\alpha$ and $\gamma$ only aggregate edges according to their direction, i.e., $\mathcal{N}$ are directed neighbors. For $\beta$, we adapt the computation of triangles to treat the entire graph as undirected. This can lead to aggregations of an edge $(u, v)$ where only the other direction $(v, u)$ is part of the graph. In this case, we aggregate over $(v, u)$, the direction that exists in the graph instead.

*Table 5.* Hyperparameter grid used on `QMD` and `QM9`.

| Hyperparameter | Values |
| --- | --- |
| Embedding dimension | 128, 256 |
| Dropout rate | 0, 0.2, 0.5 |
| Number of message passing layers | 3, 4, 5 |
| Pooling (graph-level tasks) | $\mathcal{T}_{\text{SUM}}$, $\mathcal{T}_{\text{MEAN}}$ |
| Pooling (edge-level tasks) | sum directed embeddings $\rightarrow$ make undirected prediction, make directed predictions $\rightarrow$ sum into undirected prediction |

*Table 6.* Smaller hyperparameter grid used on `QM9` after hyperparameter tuning with Tab. 5.

| Hyperparameter | Values |
| --- | --- |
| Embedding dimension | Best from Tab. 5 |
| Dropout rate | Best from Tab. 5 |
| Number of message passing layers | Best from Tab. 5 |
| Batch size | 64, 1024 |
| Pooling | $\mathcal{T}_{\text{NODESUM}}$, Best from Tab. 5 |

*Table 7.* Empirical time complexity for QM9 dataset; results from Zhou et al. (2023) and Paolino et al. (2024).

| Model | Preprocessing [sec] | Training [sec/epoch] |
|---|---|---|
| MPNN | 64 | 45.3 |
| NestedGNN | 2 354 | 107.8 |
| I2GNN | 5 287 | 209.9 |
| 2-DRFWL | 430 | 141.9 |
| 5-$\ell$GIN | 444 | 130.6 |
| EB-GNN (ours) | 100 | 31 |

*Table 8.* Average time to train different EB-GNN ablations for one epoch on our real-world datasets. All standard deviations are $\leq 0.5$s.

| EB-GNN Change | `MalnetTiny` [sec] | `QM9` [sec] | `QMD` [sec] |
|---|---|---|---|
| Unchanged | 12.4 | 17.8 | 10.7 |
| No $\beta$ aggregation | 11.6 | 16.9 | 10.5 |
| No FFN layer | 11.1 | 14.4 | 9.3 |
| No $\alpha, \beta$ aggregation | – | 14.0 | 8.5 |

# E. Speed Evaluation

**Across Models.** Similar to Zhou et al. (2023) and Paolino et al. (2024), we perform a speed evaluation on `QM9`. For this, we train our model on the training set with batch size 64 and measure the time it takes to train for a single epoch. Additionally, we also track the pre-processing time on the entire dataset. The results can be seen in Tab. 7. There are two issues with this type of evaluation. First, we (as well as previous work) compare runtime of models trained on different hardware. This is best noticed by the fact that our expressive EB-GNN is faster than the MPNN in Tab. 7. In our case, this is less of a problem because compared to 5$\ell$-GIN (the other best model on `QM9`), our GPU is significantly weaker (RTX 3080 vs RTX 3090 Ti/RTX A6000). Second, we believe that previous works included the initial processing time for the dataset in pre-processing. This includes time spent to generate the initial graphs which is needed for all models but can vary across hardware. We remedy this by reporting both the time spent for our pre-processing (70 seconds) as well as the time spent on preparing the initial graphs (30 seconds).

**Ablations.** We perform an ablation study on all our real-world datasets. We use the hyperparameter combination from our final evaluation (for `QM9` we used target $\mu$; for `QMD` the target was bond index) and computed the training time per epoch averaged over 5 epochs (all standard deviations are $\leq 0.5$s). We report the results in the Table 8. We can see that the impact of the triangle ($\beta$) aggregation is small, especially when compared to the FFN layer we use after every round of message passing. We also report the result of only using the $\gamma$ aggregation where we can see that the $\alpha, \beta$ aggregations only add a small additional overhead.

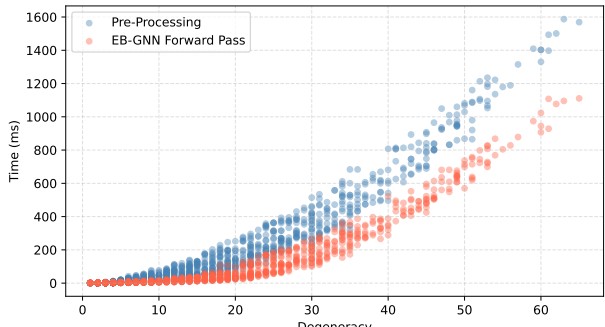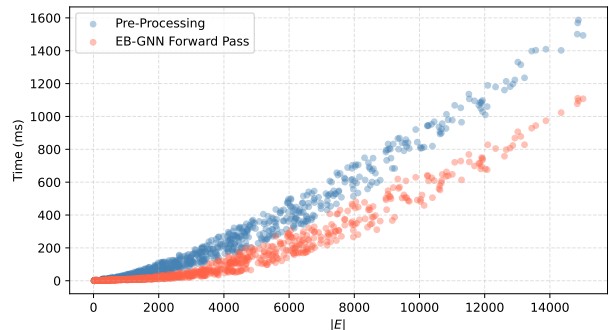

*Figure 5.* Pre-processing and inference time vs. degeneracy (left) and number of edges (right).

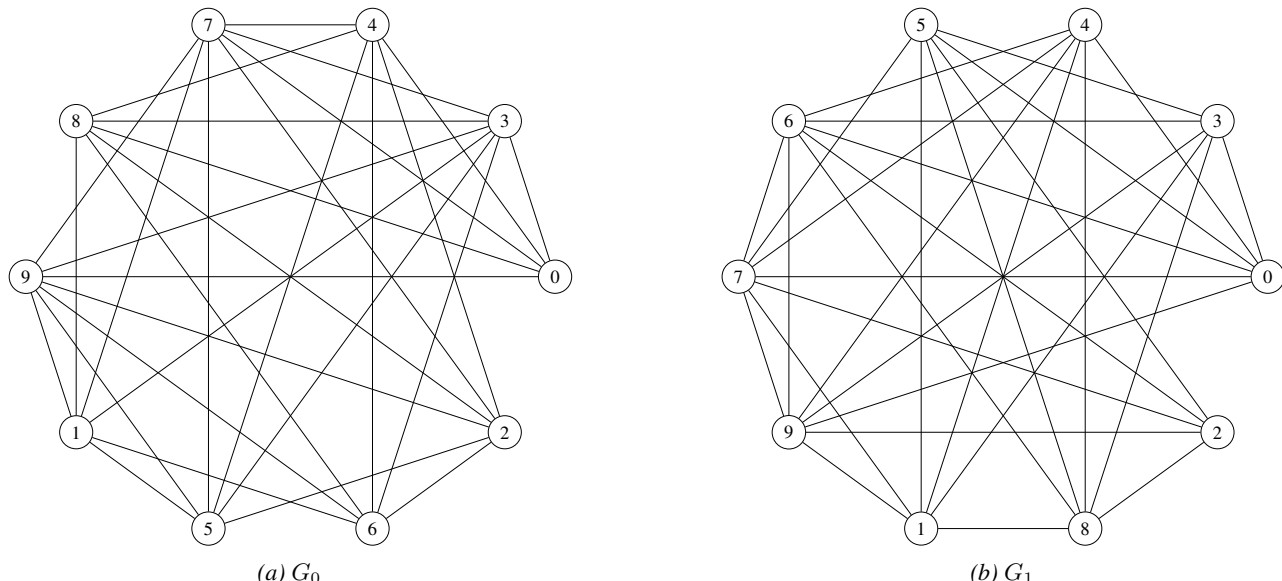

*(a) $G_0$*          *(b) $G_1$*

*Figure 6.* The first graph pair in the Extension graphs of the BREC dataset (Wang & Zhang, 2024).

**Scaling.** We empirically evaluate how the runtime of EB-GNN scales in the number of edges and the degeneracy (which upper bounds arboricity). For this, we randomly sample 1000 Erdős–Rényi graphs with the number of nodes sampled uniformly from [10, 200) and the edge probability sampled from [0.05, 0.4). For each graph we perform our pre-computation (computing triangles) and run inference with EB-GNN (batch size 1). We report the runtime of these operations averaged over 5 runs per graph. Figure 5 shows the results. Initially, the runtime increases non-linearly but then settles into an approximately linear scaling.

## F. EB-1WL vs 1WL with Triangle Information

The important role of triangles in EB-1WL naturally motivates a comparison with MPGNNs that have triangle counts injected.

Here we provide an example demonstrating that EB-1WL is strictly more expressive than 1WL with triangle counts added on node and edge level. Figure 6 shows two graphs from the BREC dataset that are indistinguishable by 1WL with such triangle counts but distinguishable by EB-1WL. Figure 7 gives example code to verify the indistinguishability by 1WL. Moreover, the graphs exhibit a different number of homomorphisms from the 4-cycle with a chord and by Theorem 4 is therefore distinguished by EB-1WL.

```python
from BRECDataset import BRECDataset
import networkx as nx

def edge_idx_to_nx(ei):
    g = nx.Graph()
    for a,b in zip(ei[0], ei[1]):
        g.add_edge(int(a), int(b))
    return g

d = BRECDataset("BREC_data_all")

g0 = edge_idx_to_nx(d[160*32*2 : (160+1)*32*2][0].edge_index)
g1 = edge_idx_to_nx(d[160*32*2 : (160+1)*32*2][1].edge_index)

def edge_tri(g, multiset=False):
    vt = nx.triangles(g)
    et = {m: vt[m[0]]+vt[m[1]] for m in g.edges()}
    if multiset:
        return list(sorted(et.values()))
    else:
        return et

nx.set_node_attributes(g0, nx.triangles(g0), 'c3')
nx.set_node_attributes(g1, nx.triangles(g1), 'c3')

nx.set_edge_attributes(g0, edge_tri(g0), 'ec3')
nx.set_edge_attributes(g1, edge_tri(g1), 'ec3')

h1 = nx.weisfeiler_lehman_graph_hash(g0, node_attr='c3', edge_attr='ec3')
h2 = nx.weisfeiler_lehman_graph_hash(g1, node_attr='c3', edge_attr='ec3')

print(h1, h2, h1 == h2)
```

*Figure 7.* Example code to verify the indistinguishability of $G_0$ and $G_1$ by 1WL with triangle counts on edges and vertices.

