# OpenReview forum: "Message Passing on the Edge: Towards Scalable and Expressive GNNs"
_ICML.cc/2026/Conference — ICML 2026 regular_

### Official Review · Reviewer_5Sbe · 2026-03-10

**Soundness:** 3
**Presentation:** 3
**Significance:** 3
**Originality:** 3
**Overall Recommendation:** 4
**Confidence:** 1

**Summary:**

This paper proposes an edge-based message passing framework that aims to go beyond the expressive limitations of standard 1WL/MPNNs while maintaining good scalability. The authors first introduce a new theoretical object, EB-1WL, which performs updates at the edge level and incorporates triangle structural information, achieving stronger discriminative power than both 1WL and NC-1WL on sparse graphs with near-linear complexity. Based on this formulation, they further design the corresponding EB-GNN architecture. The paper provides theoretical analysis of the expressive power of EB-1WL/EB-GNN and validates the method through experiments on expressiveness benchmarks, molecular property prediction, and large-scale graph classification tasks. The results suggest that the proposed approach achieves a good balance between performance and efficiency

**Compliance With Llm Reviewing Policy:**

Affirmed.

**Final Justification:**

The author has largely addressed my concerns. Considering the other reviewers' comments and the author's responses, I have decided to maintain my original score.

**Key Questions For Authors:**

please see Strengths And Weaknesses

**Limitations:**

please see Strengths And Weaknesses

**Strengths And Weaknesses:**

Strengths
1.	The method has a fairly clear novelty: By shifting message passing from the node space to the edge space and enhancing expressiveness with triangle structures, the paper explores a direction that is both natural and distinctive compared to traditional node-based MPNNs.
2.	The theoretical analysis is solid: The paper not only proves that EB-1WL is strictly stronger than 1WL and NC-1WL, but also provides a logical characterization and an expressiveness analysis related to homomorphism counting. Overall, the theoretical part is relatively complete.
3.	The complexity advantage is clear: Compared with higher-order WL/GNN methods, the proposed approach has lower complexity on sparse graphs, making it closer to practically usable scales.
4.	The experiments are fairly comprehensive: The evaluation includes both dedicated expressiveness benchmarks and real-world application tasks, such as molecular graphs and large-scale graph classification, indicating that the method is not only effective on synthetic or toy datasets designed for theoretical validation.
5.	It balances efficiency and performance well: The experiments demonstrate not only accuracy improvements but also advantages in training and inference efficiency, which strengthens the practical and engineering value of the paper.
Weaknesses
1.	The scope of applicable tasks is somewhat narrow: Since the method primarily produces edge representations, it is more naturally suited to graph-level tasks or edge-related tasks, while its support for standard node-level tasks such as node classification is relatively weak. The paper also explicitly acknowledges that it cannot directly handle general node tasks or conventional link prediction, which limits its generality.
2.	The method’s advantage depends on sparse graph structure: The complexity benefits rely on the assumption that the graph is sparse and has low arboricity. At the same time, the method involves triangle structures and edge-level interactions, so the computational cost may increase rapidly on denser graphs. In other words, its scalability advantage does not hold uniformly across all graph types.
3.	Although the expressive power is improved, it still does not reach the level of higher-order methods: EB-1WL is strictly stronger than 1WL, but it is still weaker than 2WL and cannot distinguish some more complex graph structures. Therefore, it is better viewed as an intermediate trade-off rather than a true breakthrough toward higher-order expressiveness.
4.	The comparisons with more directly related baselines could be richer: The paper already compares with methods such as NC-GNN, D-MPNN, and 5-ℓGIN, but including more modern edge-centric, line-graph-based, or triangle-aware models would make the claim of the proposed method’s advantage more convincing.

---

> ### Author Rebuttal · Authors · 2026-03-30
>
> Thank you for your review! Below we reply to the main weaknesses.
>
> > 1. The scope of applicable tasks is somewhat narrow: Since the method primarily produces edge representations, it is more naturally suited to graph-level tasks or edge-related tasks, while its support for standard node-level tasks such as node classification is relatively weak. The paper also explicitly acknowledges that it cannot directly handle general node tasks or conventional link prediction, which limits its generality.
>
>  We would like to clarify that there is no inherent limitation to using our architecture in node embedding tasks. In fact, one can use any kind of pooling over incident edges to obtain the embedding of a node. We believe that different kinds of pooling can be natural in different settings, and this should be explored in the future research.
>
>
> > 2. The method’s advantage depends on sparse graph structure: The complexity benefits rely on the assumption that the graph is sparse and has low arboricity. At the same time, the method involves triangle structures and edge-level interactions, so the computational cost may increase rapidly on denser graphs. In other words, its scalability advantage does not hold uniformly across all graph types.
>
> Our efficiency claims are intended for sparse graphs with low arboricity/degeneracy, which is the regime targeted by our method and also the regime of most real-world graph benchmarks we are aware of. For instance, the datasets in the Open Graph Benchmark (Hu et al., NeurIPS’20) and in TUDatasets (Morris et al., ICML’20 Workshops) have densities comparable to those used in our experiments. Indeed, real-world datasets from many domains are inherently sparse and have low arboricity/degeneracy (see the reference of Eppstein et al., ACM J. Exp. Algorithmics’13, and its discussion in our submission). We agree that for dense graphs, triangle enumeration and triangle-based aggregation can become substantially more expensive, so the near-linear scalability guarantees no longer apply in the same form. We are not aware of widely used benchmark datasets in this area that are substantially denser than those considered in our experiments. We will add this discussion to the limitations section of our paper.
>
> > 3. Although the expressive power is improved, it still does not reach the level of higher-order methods: EB-1WL is strictly stronger than 1WL, but it is still weaker than 2WL and cannot distinguish some more complex graph structures. Therefore, it is better viewed as an intermediate trade-off rather than a true breakthrough toward higher-order expressiveness.
>
> Let us point out that it is highly unlikely to reach the expressive power of 2WL even on sparse graphs without sacrificing scalability.  Thus, a true breakthrough toward higher-order expressiveness is probably not possible via a scalable architecture.
> The point is that the complement of a sparse graph is dense, and 2WL is able to compute the number of triangles in the complement to a graph. Computing the number of triangles in dense graphs either takes cubic time for combinatorial algorithms, or sub-cubic time but quadratic space for algorithms based on matrix multiplication.
>
> > 4. The comparisons with more directly related baselines could be richer: The paper already compares with methods such as NC-GNN, D-MPNN, and 5-ℓGIN, but including more modern edge-centric, line-graph-based, or triangle-aware models would make the claim of the proposed method’s advantage more convincing.
>
>
> For every dataset, we compare against the strongest and most recent models we are aware of. We followed the standard practice of comparing to published numbers on the datasets where those methods were evaluated. Reimplementing or retraining competing models introduces the risk of undertuning their hyperparameters. Notably, some strong models cannot be applied to other datasets. For example, 5-$\ell$GIN performs well on QM9. However, it requires us to compute paths of length 5 which is feasible on QM9 where the average graph has $18$ nodes, but infeasible on MalNetTiny where the average graph has $\geq 1500$ nodes (see table in reply to J3ww).  Our architecture frequently outperforms these baselines, and even when performance is comparable, this represents a strong result given that our architecture is significantly more runtime efficient. However, we are open for suggestion about other directly related baselines.

---

> > ### Author Rebuttal · Reviewer_5Sbe · 2026-04-01
> >
> > My concerns have been largely resolved. Taking into account the feedback from the other reviewers and the author's responses, I will keep my score unchanged.

---

### Official Review · Reviewer_AtMt · 2026-03-10

**Soundness:** 3
**Presentation:** 3
**Significance:** 2
**Originality:** 2
**Overall Recommendation:** 4
**Confidence:** 3

**Summary:**

The authors propose a novel edge-based GNN variant. The characterise its expressiveness theoretically and evaluate their method empirically.

**Compliance With Llm Reviewing Policy:**

Affirmed.

**Key Questions For Authors:**

no questions

**Limitations:**

yes

**Strengths And Weaknesses:**

This is a solid paper. The method is natural, the theoretical analysis is comprehensive, and the empirical results are good. Yet I don't think the paper meets the standards of a top-level machine learning conference like ICML. To me, there is nothing particularly innovative in this paper.

- The approach is a small variation of existing approaches. Edge-based WL is not a novel idea. There is (Liu et al. 2024). There is WL on the line graph transformation, besides (Zhang et al, 2020) and (Cai et al, 2022) also studied in (Yang, F. and Huang, X. Theoretical Insights into Line Graph Transformation on Graph Learning, 2025, https://arxiv.org/abs/2410.16138). There are higher-order WL and its restrictions, in particular restrictions to connected configurations, which for 2-WL amounts to edge-based.

I'm not saying this because I think the authors don't cover related work appropriately, but just to demonstrate that the paper's idea isn't very original. At least in my view, it is a minor variation of ideas that have been thoroughly studied.

- The theoretical results (a characterisation of the expressiveness of edge-based GNNs in terms of a restriction of 2-WL, a logical characterisation, a characterisation in terms of homomorphism counts and separation results) are carried out competently, and they are exactly what one would expect. But to me, they offer nothing particularly interesting. By now, we have seen many such characterisations. They are obtained using standard techniques that have been extensively used for related results.

- The empirical results are carried out competently on a range of typical datasets. The performance of edge-based GNNs is consistently good, though it does not stand out. Most interesting (to me) are the results for edge-level tasks, though there isn't much to compare them to here. Why, for example, is there no comparison with graph transformers on such tasks? I'd expect EB GNNs to be stronger than transformers here, but it would be good to see this.

I also wonder why the comparison against different models is not done across datasets. Specifically, transformers are only used on the security dataset and many of the GNN variants only on the molecular datasets. This looks a bit like cherry picking. In any case, it makes it hard to really get a good feeling for where the method stands for general graph learning.


Having said all this, I don't think this is a bad paper at all. I just don't think it is strong enough for ICML, because for me, there is too much routine work and too little innovation.

---

> ### Author Rebuttal · Authors · 2026-03-30
>
> Thank you for your review!
>
> >This is a solid paper. The method is natural, the theoretical analysis is comprehensive, and the empirical results are good. Yet I don't think the paper meets the standards of a top-level machine learning conference like ICML. To me, there is nothing particularly innovative in this paper.
> > The approach is a small variation of existing approaches. Edge-based WL is not a novel idea. There is (Liu et al. 2024). There is WL on the line graph transformation, besides (Zhang et al, 2020) and (Cai et al, 2022) also studied in (Yang, F. and Huang, X. Theoretical Insights into Line Graph Transformation on Graph Learning, 2025, https://arxiv.org/abs/2410.16138). There are higher-order WL and its restrictions, in particular restrictions to connected configurations, which for 2-WL amounts to edge-based.
>
> > I'm not saying this because I think the authors don't cover related work appropriately, but just to demonstrate that the paper's idea isn't very original. At least in my view, it is a minor variation of ideas that have been thoroughly studied.
> > The theoretical results (a characterisation of the expressiveness of edge-based GNNs in terms of a restriction of 2-WL, a logical characterisation, a characterisation in terms of homomorphism counts and separation results) are carried out competently, and they are exactly what one would expect. But to me, they offer nothing particularly interesting. By now, we have seen many such characterisations. They are obtained using standard techniques that have been extensively used for related results.
>
> Thank you for acknowledging that our theoretical analysis is comprehensive and that our empirical results are good. Regarding the conceptual novelty of our characterizations, we are not aware of many GNN architectures besides those corresponding to the standard $k$-WL hierarchy for which natural characterizations in terms of logic or homomorphism counts have been established previously.
>
> As a notable illustration of this fact, we note that despite message passing on line graphs being repeatedly studied in the past, no analogous results to ours have been obtained in any of these prior works. Based on this, we also believe that, while there are superficial similarities (as there are with most message-passing schemes), the specifics of the architecture that allow for such clean foundations are a major technical design challenge. Indeed, to obtain our logical characterization, we introduce a natural but new logic (see Section 4) that is exactly as expressive as our architecture.
>
> At a more general level, we consider the novelty to lie precisely in having a theoretically well-founded and empirically effective architecture that is computationally efficient enough for large graphs while improving on the severe expressivity limitations of MPNNs/1WL. In particular, the computational demands of 2WL and equivalent GNNs are known to already be prohibitive on larger graphs. At the same time, a large body of theory research in GNN theory, but also on WL processes in general, strongly suggests that the power of full $k$-WL cannot be reached without these high computational costs. In this sense, the paper’s contribution is precisely to identify and formalize a useful point in the design space that prior works did not isolate.
>
>
> > I also wonder why the comparison against different models is not done across datasets. Specifically, transformers are only used on the security dataset and many of the GNN variants only on the molecular datasets. This looks a bit like cherry picking. In any case, it makes it hard to really get a good feeling for where the method stands for general graph learning.
>
> For every dataset, we compare against the strongest and most recent models we are aware of. We followed the standard practice of comparing to published numbers on the datasets where those methods were evaluated. Reimplementing or retraining competing models introduces the risk of undertuning their hyperparameters. Notably, some strong models cannot be applied to other datasets. For example, 5-$\ell$GIN performs well on QM9. However, it requires us to compute paths of length 5 which is feasible on QM9 where the average graph has $18$ nodes, but infeasible on MalNetTiny where the average graph has $\geq 1500$ nodes (see table in reply to J3ww).  Our architecture frequently outperforms these baselines, and even when performance is comparable, this represents a strong result given that our architecture is significantly more efficient in terms of time and memory. We will clarify this explicitly in the revision and add a table indicating for each baseline whether it is inapplicable, prohibitively expensive, or simply not reported by prior work on that dataset.

---

> > ### Author Rebuttal · Reviewer_AtMt · 2026-04-01
> >
> > While my overall assessment of the paper has not fundamentally changed, reconsidering your arguments, I think that I can raise my score to "weak accept".

---

### Official Review · Reviewer_CfnQ · 2026-03-11

**Soundness:** 3
**Presentation:** 3
**Significance:** 3
**Originality:** 3
**Overall Recommendation:** 4
**Confidence:** 4

**Summary:**

This paper attempts to investigate an important aspect of graph neural network (GNN) design: improving expressivity beyond the 1-Weisfeiler–Leman (1WL) test while maintaining near-linear computational complexity. The key objective is to propose an edge-based message passing framework that captures richer structural patterns (notably triangles) without incurring the quadratic or cubic costs associated with higher-order WL methods. Building on this test, the authors propose EB-GNN, a neural architecture designed to match EB-WL expressivity. EB-GNN is evaluated on two synthetic datasets and 3 real-world datasets consisting of (a) small molecular graphs and (b) large cyber-security graphs.

**Compliance With Llm Reviewing Policy:**

Affirmed.

**Final Justification:**

Based on the recent rebuttal by the authors, I increase my score to weak accept (4).

**Key Questions For Authors:**

See weaknesses above.

**Limitations:**

yes

**Strengths And Weaknesses:**

# Strengths:

1. The paper provides a clean and well-structured theoretical framework.
2. The architecture is well aligned with the theory.
3. The key design insight is leveraging higher order interactions while keeping complexity near-linear.
4. Proposes an edge-centric GNN architecture (EB-GNN) that captures triangle structures and richer local graph patterns.

# Weakness:

1. Novelty may be incremental in the hierarchy view. From a higher-order perspective, one may view edge-based refinements like k-EB-WL as a restricted variant sitting below full (k+1)-WL. The paper should better clarify what is fundamentally new beyond a computationally cheaper restriction of higher-order WL ideas. Also, architecture novelty is weaker than theory novelty.
2. Scalability is still not fully convincing. While the asymptotic argument is appealing, the empirical evidence is mostly on selected tasks and does not fully establish how EB-GNN behaves as graphs grow in node count, edge count, or triangle density. More controlled scaling experiments would strengthen the paper.
3. Comparison with modern baselines is limited. Broader comparisons with more recent graph architectures would strengthen the empirical case, especially for the claim of being a strong general-purpose architecture.
4. Empirical gains are sometimes modest. On some benchmarks EB-GNN is competitive rather than clearly dominant, so the practical advantage over existing expressive architectures is not always decisive.
5. Comparison with higher-order/topological GNNs is missing. Since EB-GNN explicitly captures triangle interactions, it would be natural to compare against Topo-GNNs / higher-order simplicial or cell-complex GNNs. Is EB-GNN effectively a restricted instance of such models focused only on 1-simplices and triangle interactions, or does it offer something fundamentally different? This relationship is currently unclear.
6. Dense graphs are not explored. The paper’s efficiency claims rely on low arboricity and sparse-graph assumptions. For dense graphs, triangle enumeration and triangle-based aggregation may become expensive, and the scalability claims become much weaker. This needs clearer empirical evaluation and discussion.
7. Training-time cost is not fully analyzed. Although the paper gives near-linear asymptotic complexity on sparse graphs and describes preprocessing via triangle enumeration, EB-GNN introduces extra message passing through both edge neighborhoods and triangles. An ablation study on actual training/runtime overhead per layer is needed to quantify how much the triangle term contributes versus standard edge-neighbor aggregation.
8. No node classification results. The experiments only cover graph-level and edge-level tasks. Since most standard GNN benchmarks are node-level, it is unclear whether EB-GNN is suitable for node classification. The paper itself notes that EB-GNN currently produces edge embeddings only, which limits direct application to node-level tasks.

### Minor issues
Results are picked from other paper, no dataset stats are given, weak experiment section

---

> ### Author Rebuttal · Authors · 2026-03-30
>
> Thank you for your review!
>
> - **W1:** We consider the novelty to lie precisely in having a theoretically well-founded architecture that is computationally efficient enough for large graphs while improving on the severe expressivity limitations of MPNNs/1WL. In particular, the computational demands of 2WL and equivalent GNNs are known to already be prohibitive on larger graphs. At the same time, a large body of theory research in GNN Theory, but also on WL processes in general, strongly suggests that the power of full k-WL cannot be reached without these high computational costs.
>
>
> - **W2:** Thank you for raising this point. We have run an additional experiment to empirically evaluate how the runtime of EB-GNN scales; this demonstrates that the runtime scales linearly in the number of edges and the degeneracy (which upper bounds arboricity).\
> For this, we randomly sample 1000 Erdős–Rényi graphs with the number of nodes sampled uniformly from [10, 200) and the edge probability sampled from [0.05, 0.4). For each graph we perform our pre-computation (computing triangles) and run inference with EB-GNN (batch size 1). We report the runtime of these operations averaged over 5 runs per graph. The figures can be seen at [this link](https://anonymous.4open.science/r/GNN_KD_ICML-2026_Evidence-C511/README.md). As expected, the runtime of both operations scales linearly with the number of edges and the degeneracy.
>
>
> - **W3:**  For every dataset, we compare against the strongest and most recent models we are aware of, using numbers reported in the original papers. We believe that we have compared our architecture to a representative set of modern architectures. However, we are open for any suggestions for architectures to compare to.
>
> - **W4:** We agree that EB-GNN does not uniformly dominate every benchmark. What we think is important is that it achieves a strong accuracy/efficiency tradeoff: on QM9, it is best or second-best on 11/12 tasks while remaining substantially cheaper than 5-$\ell$-GIN both asymptotically and in our empirical timing study (see also Table 7 in our submission with a comparison of the preprocessing and per epoch training times); on QMD, it also yields clear gains over D-MPNN on all edge-level tasks.
>
> - **W5:** This is an interesting question. No, EB-GNNs cannot be seen as a restricted instance of topological GNNs. Indeed, topological GNNs that operate only over 1-simplices effectively lack triangle interactions. In turn, while topological GNNs over 2-simplices (triangles) do involve triangle interactions, they do it at a much larger computational cost than EB-GNNs. Indeed, message passing in such GNNs might take quadratic time even on sparse graphs with constant  arboricity (consider a graph of n triangles, having a common edge. Its arboricity is 2, but all n triangles as 2-simplices will be lower-adjacent, see [Bodnar et al, Weisfeiler and Lehman Go Topological] and thus will be sending messages each one to everybody else).
>
>
> - **W6:** Our efficiency claims are intended for sparse graphs with low arboricity/degeneracy, which is the regime targeted by our method and also the regime of most real-world graph benchmarks we are aware of. For instance, the datasets in the Open Graph Benchmark (Hu et al., NeurIPS’20) and in TUDatasets (Morris et al., ICML’20 Workshops) have densities comparable to those used in our experiments. Indeed, real-world datasets from many domains are inherently sparse and have low arboricity/degeneracy (see Eppstein et al. [ACM J. Exp. Algorithmics’13] and its discussion in our submission).
>
> - **W7:** We have performed an ablation study on all our real-world datasets. We used the hyperparameter combination from our final evaluation (for QM9 we used target $\mu$; for QMD the target was bond index) and computed the training time per epoch averaged over 5 epochs (all standard deviations are $\leq 0.5$s). We report the results in the table below. We can see that the impact of the triangle ($\beta$) aggregation is small, especially when compared to the FFN layer we use after every round of message passing. We also report the result of only using the $\gamma$ aggregation, which mimics message passing on undirected edges (no results on MalnetTiny as this dataset has directed edges). We can see that directed edge message passing only adds a small overhead compared to undirected.
>
> | EB-GNN Change | MalnetTiny | QM9 | QMD |
> | ---  | --- | --- | --- |
> | Unchanged | 12.4 | 17.8 | 10.7 |
> | No $\beta$ aggregation | 11.6 | 16.9 | 10.5 |
> | No FFN layer | 11.1 | 14.4 | 9.3 |
> | No $\alpha, \beta$ aggregation | $-$ | 14.0 | 8.5 |
>
> - **W8:**  We would like to clarify that there is no inherent limitation for using our architecture in node embedding tasks. In fact, one can use any kind of pooling over incident edges to obtain the embedding of a node. We believe that different kinds of pooling can be natural in different settings, and this should be explored in future research.

---

> > ### Author Rebuttal · Reviewer_CfnQ · 2026-04-03
> >
> > Thank you to the authors for the detailed response. However, I still have concerns regarding the following points:
> >
> > **W1.** While the theoretical contribution is appreciated, the concern remains that k-EB-WL can still be interpreted as a restricted variant of (k+1)-WL, and the novelty beyond this perspective is not fully clarified.
> >
> > **W2.** The scalability experiment is limited to very small graphs (10–200 nodes), and the plot shared does not convincingly exhibit linear scaling in more realistic regimes.
> >
> > **W4.** The empirical results still appear modest: being 1st on 3/12 and 2nd on several tasks implies it is competitive but not dominant, so I maintain my original observation.
> >
> > **W5.** My perspective that EB-GNN can be viewed as a subset/special case of Topo-GNNs still holds as authors themselves mention that Topo-GNNs pass messages over 2-simplices or triangles similar to EB-GNN.
> >
> > **W6.** I maintain my concern regarding dense graphs; as also acknowledged by the authors, performance and efficiency are likely to degrade in such settings.
> >
> > **W7.** While the increase in time from 8.5s to 10.7s (**per epoch**) may appear small, this already corresponds to **~20–25%** overhead on small graphs; such overhead is non-trivial and is likely to become more significant on larger or denser graphs.
> >
> > **W8.** EB-GNN currently does not support node classification, and given the scale of standard datasets (e.g., ogbn-products, ogbn-papers), it remains unclear how the method would scale or apply in these settings.
> >
> > Overall, due to above concerns being mostly unresolved, I stick to my original score.

---

> > > ### Author Response · Authors · 2026-04-07
> > >
> > > Thank you for your response. We appreciate the additional comments, which we will take into account to further improve our final manuscript. Below we clarify the intended claims of our paper and where we will revise the text accordingly.
> > >
> > > **W1:** In our submission we propose EB-1WL. We do not propose k-EB-WL, nor do we propose higher-order versions of our architecture. Instead, we only propose the EB-1WL test and show that it is strictly more expressive than 1WL and strictly less expressive than 2WL; we additionally give characterizations in terms of logic and homomorphism counts. Coming up with higher-order extensions may be interesting future work, but it is not part of our paper and we do not believe that generalizations are straightforward.
> > >
> > > **W2:** For the final version of our submission, we will also consider larger datasets. However, due to time constraints, we were not able to run experiments on larger graphs during the rebuttal.
> > >
> > > **W4:** We think the overall empirical picture is broader than the QM9 summary alone. Specifically, on QMD, EB-GNN is first on 6/8 tasks. On MalNet-Tiny, it is first. The comment focuses on the QM9 experiments, where we explicitly compare against much more computationally expensive architectures. For QM9, we agree that EB-GNN is competitive rather than dominant, as explicitly stated in the submission: “while remaining competitive with more expressive models, at a substantially lower runtime” (Line 434). Thus, our contribution is to provide an architecture with strong accuracy–efficiency trade-offs.
> > >
> > > **W5:** As we highlighted in our response above, we agree that Topo-GNNs are somewhat similar in spirit but they do not subsume our concrete developments. Specifically, a key difference is the computational efficiency of the two architectures. In the example above, we showed that if all triangles share a single edge, Topo-GNNs require quadratic time, whereas our EB-GNN only requires linear time. Therefore, even though there are some similarities in the message-passing design, there can be substantial differences in the computational efficiency. Thus, our contribution is an architecture/test with specific sparse-graph guarantees that generic Topo-GNN architectures do not provide.
> > >
> > >
> > > **W6:**
> > > > I maintain my concern regarding dense graphs; as also acknowledged by the authors, performance and efficiency are likely to degrade in such settings.
> > >
> > >  Please note that we never claimed that predictive performance is likely to degrade on dense graphs, nor is there any evidence for this claim. However, as the reviewer has noted, the runtime of our model does increase on dense graphs. However, this is not a major limitation for two reasons.
> > >
> > > - (1) Real-world graphs are typically sparse. Empirical studies (e.g., Eppstein et al., 2013) show that even very large graphs have relatively low arboricity. This is consistent with our datasets, where arboricity is at most 15 (Line 295). Thus, the dense regime is rarely encountered in practice.
> > >
> > > - (2)  in dense graphs, $\alpha$ approaches the number of nodes $\sqrt{m}$. $O(m\sqrt{m})$ runtime is considered the hard theoretical limit for counting triangles in a graph. Hence, the runtime of EB-GNN is close to algorithmically optimal which means that every other GNN that is strictly more expressive cannot be significantly more efficient than EB-GNN.
> > >
> > > **W7:** We agree that the triangle term introduces nontrivial overhead. Our claim is not that this overhead is negligible in all regimes, but that on our current benchmarks it is moderate relative to the overall training cost; we will revise the wording accordingly.
> > >
> > > **W8:** We agree that the current paper does not present node-classification experiments. However, we would like to stress that EB-GNN supports node-level tasks: just as one classically pools node embeddings for graph tasks, one can pool embeddings of incident edges for node tasks. What we write about this in our “Limitations” discussion (starting Line 414, right column) may be overly cautious, but it is intended to convey that detailed behaviour of such pooling approaches merits its own deeper investigation.

---

### Official Review · Reviewer_J3ww · 2026-03-12

**Soundness:** 3
**Presentation:** 4
**Significance:** 3
**Originality:** 3
**Overall Recommendation:** 5
**Confidence:** 3

**Summary:**

This paper introduces EB-GNN, a GNN based method that perform message passing on edge. EB-GNN allows to increase expressive power while keeping the computational cost low. First of all the authors propose the EB-WL test, i.e., an edge-based color refinement test; then they propose EB-GNN which mimics the EB-WL test. The authors provide theoretical results showing that EB-WL is more expressive than the standard WL test and that EB-GNN matches its expressive power. They also provide a logical characterization and connections with homomorphism counting. In addition, the paper analyzes the computational complexity of the proposed method and shows that it can run in near-linear time on sparse graphs. the model is evaluated on synthetic benchmarks designed to test expressivity and on several real-world datasets, including molecular prediction tasks, even at edge level.

**Compliance With Llm Reviewing Policy:**

Affirmed.

**Final Justification:**

I maintain my initial positive score. Given the quality of the article and the clarifications received in the reply, I confirm that, in my opinion, the article deserves acceptance.

**Key Questions For Authors:**

see weaknesses.

**Limitations:**

yes

**Strengths And Weaknesses:**

**Strengths**

The paper is well written and easy to follow. The theoretical section is very solid, and the notation is consistent.The way in which EB-GNN works is very simple and intuitive, which is a strong point to me, since it is not an heavily engineered model. I really  appreciate the goal of designing a model that is both expressive and computationally efficient. This is an important direction for the GNN community, since many expressive architectures become impractical due to their high computational cost. The near-linear complexity of the proposed method is a very appealing aspect of the model. Another aspect that I found interesting is the fact that you can easily use EB-GNN for edge representation task, which I think is good since usually the expressivity is very much studied in terms of graph-level, very few in terms of node or edge level. Finally, the analysis of the computational complexity is convincing. The lower computational cost of the proposed method compared to higher-order GNNs is an important contribution of the paper.

**Weaknesses**

The main weakness concerns the discussion of expressivity in relation to practical cost and generalization.
Recently, the field of GNN expressivity has been moving toward a better understanding of when higher expressivity is actually needed and at what cost, both in terms of generalization and computational complexity (see for example [1,2,3,4]). For this reason, I believe that a paper that focuses on expressivity should ideally also discuss these aspects more directly.

1) Regarding generalization, the authors explicitly mention in the limitations that the paper does not include an analysis of generalization. I understand that a single paper cannot contains every aspect of the problem. It is reasonable to leave some directions for future work. However, given how the field of expressivity is currently evolving, I feel that expressivity and generalization are becoming increasingly intertwined topics. Because of this, it might have strengthened the paper to include at least a small analysis or discussion about generalization behavior when using a more expressive architecture. Even a limited empirical exploration could have added useful insights.

2) Regarding computational cost, the authors provide a detailed theoretical analysis showing that the method has lower complexity than many expressive alternatives. I found this analysis very nice. However, it would also have been helpful to see a more practical view of the computational cost. For example, reporting the runtime per epoch would give a clearer picture of the actual computational advantage. While the asymptotic complexity is formal and good to see, big-O notation can sometimes be hard to grasp, and without knowing values such as the maximum degree or number of edges it can be difficult to estimate the real difference in practice.

I have some questions:
1. do the authors have an intuition about why the method fails on the CFI graphs?
2. since the model is been applied also to edge representation tasks, do the authors have some intuition about expressiveness of EB-GNN with respect to other link-representation methods? there are a bunch of works analyzing expressiveness in terms of capability of producing different representations for not automorphic links (see for example [5,6]) and I am wondering how EB-GNN behave regarding this.

[1] Fabian, Pascal Welke, and Thomas Gärtner. "Is Expressivity Essential for the Predictive Performance of Graph Neural Networks?." NeurIPS 2024 Workshop on Scientific Methods for Understanding Deep Learning. 2024.

[2] Franks, Billy Joe, et al. "Weisfeiler-Leman at the margin: When more expressivity matters." Forty-first International Conference on Machine Learning.

[3] Maskey, Sohir, et al. "Graph Representational Learning: When Does More Expressivity Hurt Generalization?." arXiv preprint arXiv:2505.11298 (2025).

[4] Carrasco, Martin, et al. "Rademacher Meets Colors: More Expressivity, but at What Cost?." arXiv preprint arXiv:2510.10101 (2025).

[5] Lachi, Veronica, et al. "Bridging Theory and Practice in Link Representation with Graph Neural Networks." The Thirty-ninth Annual Conference on Neural Information Processing Systems.

[6] Zhang, Muhan, and Yixin Chen. "Link prediction based on graph neural networks." Advances in neural information processing systems 31 (2018).

---

> ### Author Rebuttal · Authors · 2026-03-30
>
> Thank you for your positive review. Below we respond to the weaknesses and questions.
>
> **Relation to generalization:**
> From our point of view, there are two central problems when it comes to expressivity and generalization:
> - (I): Expressivity that is too high and misaligned with the task can lead to worse generalization than using a less expressive architecture (see Proposition 14 in [2] and Theorem 5.1 in [3]).
> - (II): Highly expressive models that achieve good generalization might do so at the cost of significant runtime without actually needing such high expressivity [1].
>
> Our architecture is less impacted by both issues: its near-linear runtime bounds the effective expressivity, reducing the risk of task misalignment (I) and removes  the motivation for sacrificing expressivity in favor of speed (II).  We will add a discussion about this to the final version of our paper.
>
> [1] Fabian, Pascal Welke, and Thomas Gärtner. "Is Expressivity Essential for the Predictive Performance of Graph Neural Networks?." NeurIPS 2024 Workshop on Scientific Methods for Understanding Deep Learning. 2024.
> [2] Franks, Billy Joe, et al. "Weisfeiler-Leman at the margin: When more expressivity matters." Forty-first International Conference on Machine Learning.
> [3] Maskey, Sohir, et al. "Graph Representational Learning: When Does More Expressivity Hurt Generalization?." arXiv preprint arXiv:2505.11298 (2025).
>
> **For generalization analysis:** Do you have any suggestions for small experiments that could allow us to empirically analyze generalization properties?
>
> **For computational cost:** We have performed the following additional experiments to better investigate how our architecture scales:
> - We plotted the running time of EB-GNN against different graph parameters and show that our architecture scales linearly in the number of edges and the node degeneracy (you can find plots under the link https://anonymous.4open.science/r/GNN_KD_ICML-2026_Evidence-C511/README.md). We give further details in the response to Reviewer CfnQ.
> - Additionally, we have performed an ablation study of different components of our architecture. See the response to Reviewer CfnQ for details.
> - Finally, below we give an overview of the inference speed of EB-GNN with different structural parameters of our datasets (where we note that degeneracy is an upper bound on arboricity, which we have already reported in our paper).
>
> In the table below, we report the time per epoch on different datasets. We can see that on all datasets training a single epoch takes approximately the same time (models use hyperparameters from our evaluation). However, even though MalNetTiny contains only 5000 graphs, one epoch is slower than a single epoch on 65 000 graphs from QMD. This is because graphs in MalNetTiny are both larger (1415 edges vs 20 edges) and denser (average degeneracy 15 vs 3).
> | Dataset | #Graphs | Avg. #Edges | Avg. #Nodes | Max Degree | Max Degeneracy | Avg. Degeneracy | EB-GNN Training Speed in s/Epoch
> | ---  | --- | --- | --- | --- | --- | --- | --- |
> | MalNetTiny | 5 000 | 1415 | 1505 | 433 | 15 | 4 | 12.4s |
> | QM9 | 130 000 | 19 | 18 | 5 | 3 | 2 | 17.8s  |
> | QMD | 65 000 | 20 | 20 | 4 | 3 | 2 | 10.7s |
>
> **Question 1:** Yes, the CFI graph dataset in BREC primarily consists of a rather specific type of 1-WL indistinguishable graph pairs. Specifically, many of the graphs in the CFI part of the dataset are of the form $C_k+C_k$ and compared against $C_{2k}$, i.e., the disjoint union of two $k$-node cycles needs to be distinguished from a cycle with $2k$ nodes. This is a rather limited view of the limitations of 1-WL expressivity and, crucially, for $k>3$ it does not include any triangles that our architecture could exploit.
>
> We will report all of these additional results in the final version of the paper.

---

> > ### Author Rebuttal · Reviewer_J3ww · 2026-04-03
> >
> > I would like to thank the authors for their detailed responses to my comments. I do not have any further questions or suggestions. I maintain my original score, which was already positive, and I believe it appropriately reflects the quality of the paper.

---

### Decision · Program_Chairs · 2026-04-30

**Decision:**

Accept (regular)

**Comment:**

**Overall Assessment**

This paper received consistently positive reviews and presents a strong, well-motivated contribution to the GNN literature. It introduces a principled edge-centric framework that improves expressivity with respect to standard MPNNs while aiming to preserve computational efficiency. The work is theoretically solid, clearly written, and addresses an important challenge in GNN design: balancing expressiveness with scalability.

**Summary of Contributions**

The paper proposes EB-GNN, an edge-based message passing architecture inspired by a new edge-based Weisfeiler–Lehman test (EB-WL). The authors first formalize EB-WL as an edge-centric color refinement procedure and prove that it is strictly more expressive than the standard 1-WL test. They then design EB-GNN to mirror the discriminative power of EB-WL, with theoretical guarantees that the model matches this expressivity. Beyond expressiveness, the paper provides a first-order logic characterization of EB-1WL, connections to homomorphism counts, and a detailed complexity analysis showing near-linear runtime on sparse graphs with low arboricity. Empirically, EB-GNN is evaluated on both synthetic benchmarks designed to probe expressivity and several real-world datasets, including molecular prediction tasks at both graph and edge levels. The results demonstrate competitive or improved performance while maintaining reasonable computational costs.

**Strengths**

> The paper is well written, clearly structured, and easy to follow.

> The expressivity analysis, WL comparisons, and logical characterizations are rigorous and convincing.

> EB-GNN is conceptually simple and intuitive, avoiding heavy engineering while being theoretically grounded.

> The focus on achieving higher expressivity without prohibitive computational cost is highly relevant, as many expressive GNNs suffer from poor scalability in practice.

**Weaknesses**

> While the asymptotic analysis is appealing, empirical scalability experiments are limited. The paper does not fully demonstrate how EB-GNN behaves as node count, edge count, or triangle density increases in a controlled setting.

> Comparisons with more recent or competitive expressive GNN architectures are sparse.

> The efficiency claims hinge on sparsity and low arboricity assumptions. For dense graphs, the reliance on triangle enumeration and triangle-based aggregation may significantly weaken scalability.

> Although preprocessing and asymptotic complexity are discussed, there is no ablation or runtime breakdown quantifying the overhead introduced by triangle-based message passing compared to standard edge-neighbor aggregation.

> Experiments are restricted to graph-level and edge-level tasks. Node classification results would improve the completeness of the empirical evaluation.

Despite the above weaknesses, the paper makes a clear and valuable contribution, especially its theoretical depth, clean model design, and focus on practical expressivity.